


# Lagrangian tracking of sea ice in Community Ice CodE (CICE; version 5)

Chenhui Ning[1], Shiming Xu[1,2], Yan Zhang[3,1], Xuantong Wang[1], Zhihao Fan[1], and Jiping Liu[4,5]

[1]Ministry of Education Key Laboratory for Earth System Modeling, Department of Earth System Science (DESS), Tsinghua University, Beijing, China
[2]University Corporation for Polar Research (UCPR), Beijing, China
[3]Institute of Applied Physics and Computational Mathematics (IAPCM), Beijing, China
[4]School of Atmospheric Sciences, Sun Yat-sen University, Zhuhai, China
[5]Southern Marine Science and Engineering Guangdong Laboratory (Zhuhai), Zhuhai, China

**Correspondence:** Shiming Xu (xusm@tsinghua.edu.cn) and Jiping Liu (liujp63@mail.sysu.edu.cn)

**Abstract.** Sea ice models are essential tools for simulating the thermodynamic and dynamic processes of the sea ice and the coupling with the polar atmosphere and ocean. Popular models such as the Community Ice CodE (CICE) are usually based on non-moving, locally orthogonal Eulerian grids. However, the various in-situ observations such as ice tethered buoys and drift stations, are subjected to sea ice drift and hence by nature Lagrangian. Furthermore, the statistical analysis of sea ice

kinematics requires the Lagrangian perspective. As a result, the offline sea ice tracking with model output is usually carried out for many scientific and validational practices. Certain limitations exist, such as the need of high frequency model outputs, as well unaccountable tracking errors. In order to facilitate Lagrangian diagnostics in current sea ice models, we design and implement an online Lagrangian tracking module in CICE under the coupled model system of CESM (Community Earth System Model). In this work, we introduce its design and implementation in detail, as well as the numerical experiments for

the validation and the analysis of sea ice deformations. In particular, the sea ice model is forced with historical atmospheric reanalysis data and the Lagrangian tracking results are compared with the observed buoys' tracks for the years from 1979 to 2001. Moreover, high-resolution simulations are carried out with the Lagrangian tracking to study the multi-scale sea ice deformations modeled by CICE. Through scaling analysis, we show that CICE simulates multi-fractal sea ice deformations in both the spatial and the temporal domain, as well as the spatial-temporal coupling characteristics. The analysis with model

output on the Eulerian grid shows systematic difference with the Lagrangian tracking-based results, highlighting the importance of the Lagrangian perspective for scaling analysis. Related topics, including the subdaily sea ice kinematics and the potential application of the Lagrangian tracking module, are also discussed.



# 1 Introduction

Sea ice floes are inherently Lagrangian points that undergo thermodynamic and dynamic changes throughout the lifetime. Under the dynamic forcings from the atmosphere and the ocean, the sea ice drifts and with the build-up of internal stress, deforms and undergoes plastic failures. The drift of sea ice floes is associated with constant thermodynamic growth and melt of the sea ice, hence fundamental to the energy and ice/water balance in the polar regions (Haas, 2009). Furthermore, highly nonlinear and anisotropic linear kinematic features manifest with the sea ice deformations, which are prevalent from meters to

the geophysical scales (Marsan et al., 2004). The accurate observation and modeling of the sea ice drift and deformations are key to both our scientific understanding of the climate system and human activities in the polar region.

Due to the harsh conditions of the polar environment, the long-term direct measurements of the sea ice are usually carried out through in-situ deployments of buoys. These autonomous systems, which are usually attached to the sea ice, report back their locations, the sea ice conditions, as well as the associated atmospheric and ocean conditions. Since they drift with the

sea ice, their locations are also representative of the sea ice floe they are attached to. For example, the International Arctic Buoy Programme (IABP: https://iabp.apl.uw.edu) compiles historical and real-time buoy measurements in the Arctic region. The data are widely used in the study of sea ice dynamic (Rigor et al., 2002) and thermodynamic processes (Perovich et al., 2014), as well as the data assimilation for numerical weather forecasts (Inoue et al., 2009).

The drift and the deformation of sea ice are the result of its dynamic response to the atmospheric and oceanic forcings.

Unlike the Newtonian fluids of the air and the water, the sea ice patches undergo deformations with distinctive characteristics of multi-fractal, plastic faults. These deformations are highly localized and anisotropic, which usually correspond to sea ice lead and ridges and manifest as linear kinematic features [LKFs, (Kwok et al., 1998)]. While sea ice ridging is the major way of forming thick sea ice, leads are important source of heat and moisture for the polar regions especially during winter (Rothrock, 1975; Andreas and Cash, 1999). Therefore, sea ice deformations are crucially important for the coupled climate

system of the polar region. In order to study how the sea ice deforms, we usually carry out multi-scale analysis through the statistics of the deformation rates (i.e., the speed at which the sea ice deforms). Specifically, the deformation rates, denoted $\dot{\varepsilon}$'s, can be computed for individual sea ice patches at various temporal and spatial scales. Since the sea ice is constantly drifting, the computing of $\dot{\varepsilon}$'s and the related analysis should also be carried out from a Lagrangian perspective. Satellite remote sensing based datasets such as RGPS (Lindsay and Stern, 2003) utilize SAR images collected at different times to produce large-scale,

high-resolution maps of the sea ice drift and deformation. In particular, both the correlation-based and the feature tracking approaches when processing the SAR images ensure that the analysis of the drift/deformation is Lagrangian by nature. These datasets are widely used for the study the multi-scale sea ice kinematics (Marsan et al., 2004) and the validation of numerical simulations (Kwok et al., 2008; Rampal et al., 2019).

For the simulation of sea ice and its kinematics, we construct numerical models which involves a layered structure with: (1)

the mathematical modeling of the physical processes, (2) the numerical treatments for the spatial-temporal discretization and the integration, and (3) the code implementation and the simulation on parallel computers. Popular sea ice models, such as CICE (Community ICE code, https://github.com/CICE-Consortium/CICE) and SI[3] (Vancoppenolle et al., 2023), are usually based on



the spatial discretization on locally orthogonal structured grids. With the rheology model such as Viscous-Plastic [VP, see also Elasto-Viscous-Plastic (Hunke and Lipscomb, 2008)], these models (Bouchat et al., 2022; Hutter and Losch, 2020) are capable to reproduce certain statistics of the observed multi-scale sea ice kinematics (Kwok et al., 2008). However, the model's output, including the instantaneous and the average model status at different temporal scales, are typically defined on the model's native, Eulerian grid. One notable exception is the neXtSIM model, which is based on Lagrangian moving mesh and natively supports the scaling analysis (Rampal et al., 2019). But for CICE and many widely adopted sea ice models, the model output is insufficient especially for the scaling analysis at large temporal scales, since it inherently requires a Lagrangian perspective. A typical practice to overcome this limitation is to reconstruct Lagrangian tracks with the model's output, such as that based on the models' daily velocity fields in Bouchat et al. (2022). Certain limitations are still present, however, especially given the ever-growing resolution of current models (Xu et al., 2021; Zhang et al., 2023). High-frequency model output is needed to reconstruct realistic Lagrangian tracks, which requires large amount for data storage and off-line computation. Furthermore, the analysis of small-scale sea ice deformations [i.e., minute-scale as in Oikkonen et al. (2017)] requires even finer spatial and temporal model output and larger overhead with the offline tracking analysis. Therefore, more flexible Lagrangian diagnostic tools are needed for the scaling analysis of sea ice kinematics and future development of sea ice models.

In this paper we introduce the online Lagrangian tracking of sea ice and its model integration in the model of CICE (version 5). The tracking of sea ice is carried out along with the model's numerical integration, and it supports very high-frequency tracking (at the model's time step) and large numbers of Lagrangian points. The model integration is carried out and further validated through the numerical experiments in the coupled framework of Community Earth System Model (CESM, version 2: https://www.cesm.ucar.edu/models/cesm2). Specifically, the comparison with observed buoy tracks is carried out with atmospherically forced historical simulations. Furthermore, high-resolution experiment with $7km$-resolution in the Arctic region is carried out, and we evaluate the spatial-temporal scaling of wintertime sea ice deformations. In Section 2 we introduce in detail the tracking algorithm and the integration in CICE. Section 3 includes the numerical experiments and detailed analysis of simulation results. Finally in Section 4, we summarize the article and discuss related topics, including potential applications of the sea ice Lagrangian tracking and the high-resolution simulation of sea ice kinematics.

## 2   Lagrangian tracking in CICE

The Lagrangian tracking of sea ice is tightly integrated with the dynamics processes of CICE (version 5). The model grid of CICE is a two-dimensional, logically rectangular, structured grid with the size of (nx_global,ny_global) and indexed by (i,j), respectively. Typical lateral boundary conditions are supported for global configurations, including the east-west periodic boundary and the tripolar grid boundary (Murray, 1996). For the Lagrangian tracking, each active point has a specific logical location of $(x,y)$ in the two-dimensional continuous space, satisfying: $0 \leq x <$ nx_global and $0 \leq y <$ ny_global. Furthermore, there is a bi-projection between the point's logical location and the corresponding geolocation. The Lagrangian tracking is carried out within the model grid's domain following the tracking algorithm, while maintaining the grid's lateral boundary conditions and the land/ocean distribution.





In this section, the Lagrangian tracking algorithm is introduced in detail in Sec. 2.1, which is based on the existing advection framework of transport remapping (Dukowicz and Baumgardner, 2000). Regarding the commonly used functionality of CICE, the Lagrangian tracking also supports parallel computing, tripolar grids, and a simple logging system for the record of the tracking results. Details implementation and the integration in CICE is covered in Sec. 2.2.

## 2.1 Lagrangian tracking algorithm

The Lagrangian tracking algorithm is based on the transport remapping advection scheme, which is available in CICE (Dukowicz and Baumgardner, 2000). As a conservative, two-dimensional, semi-Lagrangian scheme, transport remapping is based on the Arakawa-B staggered grid. At each dynamics step of CICE, the backward tracking of the corner points is carried out for the advection of tracers onto the Eulerian grid cells (corners of the hollow quadrilateral in Fig. 7.a). For the Lagrangian tracking, we utilize the backward tracking vectors and compute the forward tracking for all 4 corner points (filled grey arrows in Fig. 7.a), and they form the new quadrilateral (outlined in blue). Then based on the previous location of the Lagrangian point (black circle), we can compute its new location (blue circle) as follows.

The tracking of the Lagrangian point is based on the local, normalized coordinate of the grid cell that the point is in. Under the local coordinate, the cell's four corners corresponding to $(0.0, 0.0)$, $(0.0, 1.0)$, $(1.0, 0.0)$ and $(1.0, 1.0)$ respectively. The previous location of the Lagrangian point under this coordinate is $(x_{local}, y_{local})$, satisfying: $0 \leq x_{local} < 1$ and $0 \leq y_{local} < 1$. Note that given the point's logical position in the model grid as $(x, y)$, the following hold: (1) the cell the point is present has the index of $(\lfloor x \rfloor + 1, \lfloor y \rfloor + 1)$; and (2) $x_{local} = x - \lfloor x \rfloor$ and $y_{local} = y - \lfloor y \rfloor$. After the forward tracking, the four corners have the new locations of: $(x_{i,j}, y_{i,j})$ for $i \in \{0, 1\}$ and $j \in \{0, 1\}$. Then, the new location of the Lagrangian point, denoted $(x^*_{local}, y^*_{local})$ is computed through the bilinear interpolations with $x_{i,j}$'s and $y_{i,j}$'s, assuming that its relative location within the new quadrilateral remains $(x_{local}, y_{local})$. In the case that $x^*_{local}$ or $y^*_{local}$ is larger than 1 or smaller than 0, the Lagrangian point has drifted out of the current grid cell, which we denote as the migration of the point. Since CICE utilizes domain decomposition for parallel computing, the migration potentially is between blocks (sub-domains after decomposition) and entails communication between parallel processes. The detailed support is introduced further in Section 2.2.

Based on the geolocations of the four cell corners and the Lagrangian point's relative location within the cell, we can compute the geolocation of the Lagrangian point. Specifically, first, the three-dimensional locations of the cell corners are computed with their latitudes and longitudes. Through linear interpolations, we locate the three-dimensional location of the Lagrangian point. Finally, the latitude and longitude of the Lagrangian point can then be determined.

## 2.2 Implementation in CICE

### 2.2.1 Basic support

For the Lagrangian tracking in CICE, we define the data structure is `lagr_point` in the FORTRAN module of `ice_transport_driver`. It contains necessary fields of information for the Lagrangian point, including its current status, its lifetime, its current location, and other essential information. Furthermore, each of the parallel CICE processes contains a large, pre-allocated pool of avail-

(a) Backward and forward tracking of U/V points

(b) Mapping of the Lagrangian point

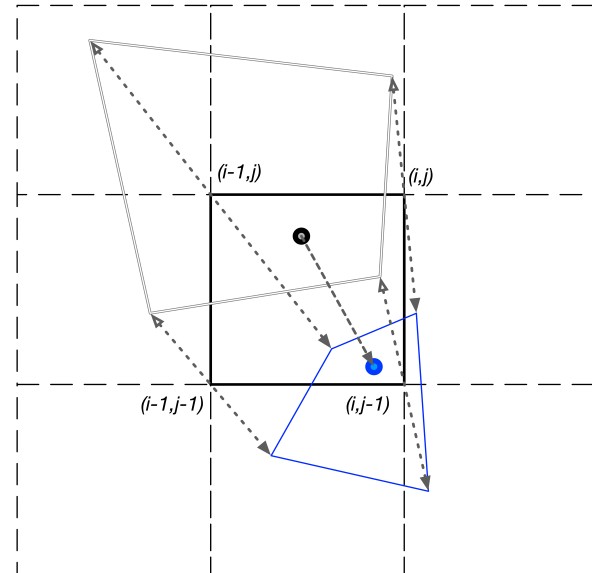
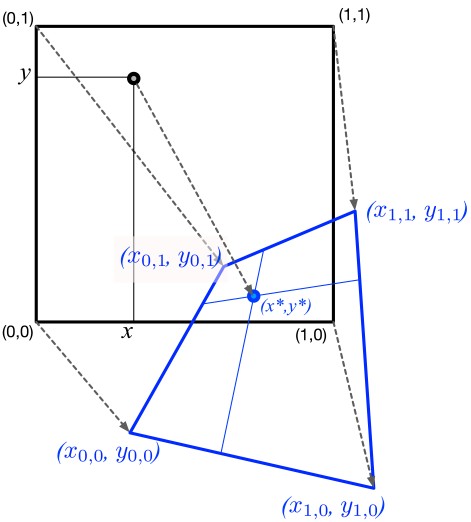

**Figure 1.** Local Lagrangian tracking scheme. Panel (a) shows the backward tracking (the double-lined quadrilateral) and the forward tracking (the quadrilateral in blue line) of the corners of the current T-cell at $(i, j)$ (thick black quadrilateral). In panel (b), the tracking of a Lagrangian point (black circle) at normalized relative position $(x, y)$ within the current T-cell is based on linear interpolation of the four corner points [i.e., $(x_i, y_j)$, for $i = \{0, 1\}$ and $j = \{0, 1\}$].

able Lagrangian points (i.e., instances of `lagr_point`). When a new Lagrangian point is created in the current process or migrated from another process, an unused slot is claimed from this pool. Similarly, when the point is dead (i.e., due to melting) or migrating out of the block, the slot is reclaimed and recycled in the pool.

The life cycle of a Lagrangian point consists of several stages (Fig. 2) and the transition between the stages, called events. Upon its creation (type-I event), the Lagrangian point is assigned to a specific geolocation, and consequently, a specific block and a specific processor. The Lagrangian point drifts (type-II event), until the sea ice melts (sea ice concentration lower than 5%, type-V event) or it is automatically deactivated (e.g., exceeding the prescribed maximum lifetime). When the Lagrangian points migrates outside the current block, it will be delivered to the corresponding adjacent block (type-III event, also see below). This process is carried out through the built-in boundary exchange operations of CICE. A special *landing* event of Lagrangian points (type-IV) is supported, but not possible in the current implementation.

### 2.2.2 Overall model integration

As shown in Figure 3, the tracking of all Lagrangian points occurs within each dynamics time step (i.e., `step_dynamics`) after the computation of the prognostic velocities and the advection process (which computes the backward tracking vectors).



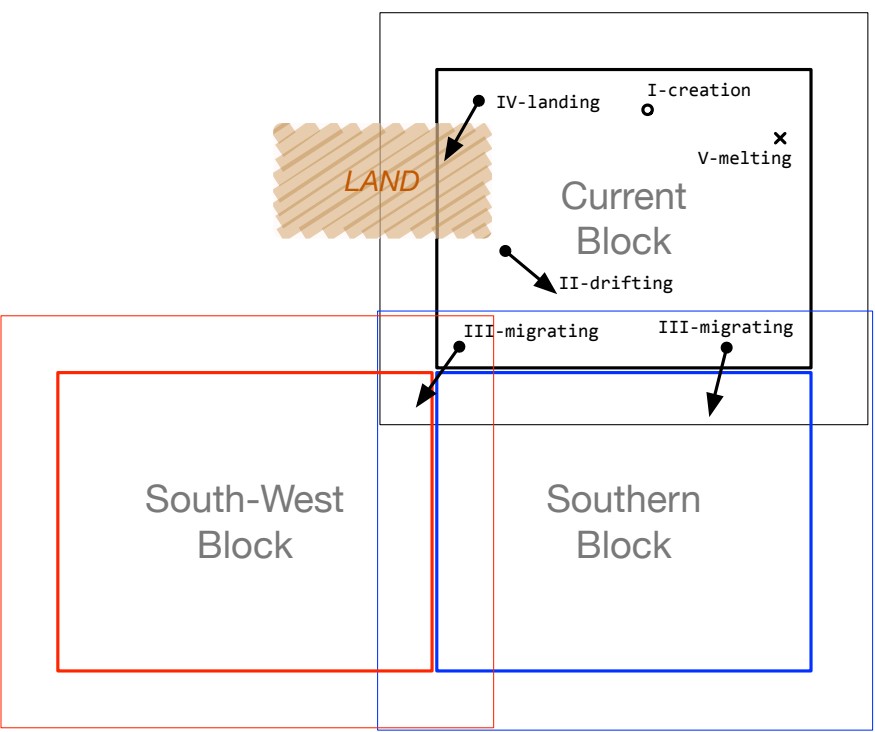

**Figure 2.** Typical events for Lagrangian points, including: the creation, the melt (i.e., ice concentration below 5%), drifting within a block or between blocks (i.e., migration), and migration onto lands.

To ensure numerically stable integration for the advection, there is strict limitation over the time step and hence the backward/-forward tracking vectors. As a consequence, Lagrangian points cannot migrate beyond one grid cell in either direction. When a specific point migrates beyond the current block's boundary (i.e., thick rectangular boundary in Fig. 2), it will be strictly within the outer boundary of the block (i.e., within the thin rectangular boundary in Fig. 2).

The Lagrangian tracking contains 4 major steps (shown below). For each Lagrangian point, its status is maintained, tracked with the forward drifting vectors, and the location information updated. In the case of the Lagrangian point migrates outside the current block, it is recorded for further boundary exchange. Then the Lagrangian points are exchanged between blocks, with newly migrated Lagrangian points recorded for the current block.

1. For each active Lagrangian point:

(a) Check for deactivation (due to melt, life-span, etc.), with necessary management of the Lagrangian point slots

(b) Aging of the Lagrangian point

(c) Retrieve the local advection information





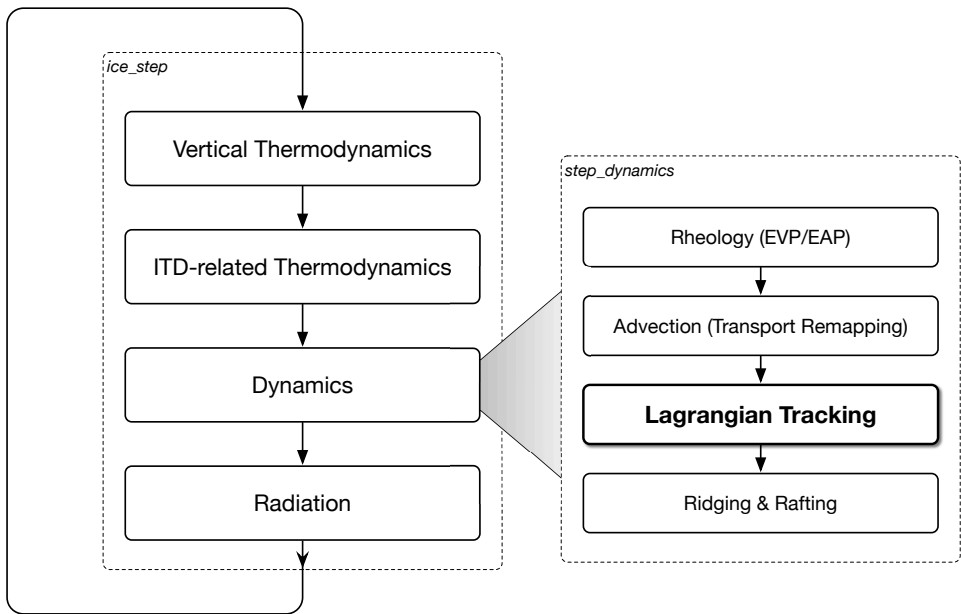

**Figure 3.** Overall integration of Lagrangian tracking in the time step of CICE.

   (d) Do Lagrangian tracking, and update the logical and the geophysical positions

   (e) Check for potential migration out of the current block

(f) For a migrating point:

      i. Record its information for boundary exchange

      ii. Carry out management of the Lagrangian point slot

2. Boundary exchange for Lagrangian points

3. Activate newly migrated Lagrangian points

4. Deactivate Lagrangian points (due to landing, etc.) with necessary managements

### 2.2.3 Miscellaneous

**Boundary exchanges for migrating Lagrangian points:**

Each dynamics time step involves a single FORTRAN function call to the boundary exchange, carrying out the migration of Lagrangian points between blocks. The maximum exchangeable Lagrangian point count per boundary cell can be changed





in the code through the FORTRAN parameter `LAGR_BNDY_SIZE_PARAM`. It is worth to note that the single boundary exchange due to the migration of Lagrangian points only incurs a very small computational overhead. Within the same dynamics time step, the EVP solver usually contains over 100 subcycles of the elastic waves and the corresponding boundary exchanges (Lemieux et al., 2010; Xu et al., 2021). Furthermore, the Lagrangian tracking utilizes the transport remapping scheme's backward tracking. In general, the EVP solver dominates the overall simulation time of the sea ice dynamics, with the Lagrangian

tracking consuming less than 3% of the time for all the numerical experiments in Section 3.

**Support for tripolar grids:**

Tripolar grids are commonly used in high-resolution global configurations of CICE and CESM (Small et al., 2014). For example, the 0.1° grid of TX0.1 is used for the global ocean mesoscale-resolving simulations of CESM. The commonly used U-fold tripolar grids are supported for our implementation of the Lagrangian tracking in CICE, with the schematics shown in

Fig. 4.

When the Lagrangian point drifts beyond the northern boundary of the blocks on the tripolar folding line, it is passed to the corresponding block and grid cell through the boundary exchange process. This corresponds to the case with $y$ larger than `ny_global`, where `ny_global` is the grid cell count in the $j$ direction. Then, the $x$ and $y$ position of the point are modified accordingly to the correct values, following the topology on the tripolar boundary.

**Logging system:**

A simple text-based logging system is implemented to report the results of the Lagrangian tracking. Each parallel processor generates a log file that contains the records of all active Lagrangian points on the processor. The records include the major events, including the creation of the Lagrangian points, the melt events, the migration events, as well as the Lagrangian points' status at regular time intervals. The time step count for reporting Lagrangian points' information is a user-prescribed compile-

time parameter (see Appendix C for details).

### 3 Numerical experiments and analysis

We carry out numerical experiments of the Lagrangian tracking with CICE (version 5) and CESM (version 2). CICE is configured with 5 ice thickness categories with multiple vertical layers, as well as full thermodynamic and dynamic processes. Key model configurations include the mushy-layer vertical thermodynamics, the Elasto-Viscous-Plastic (EVP) rheology model, and

the Delta-Eddington radiation scheme. Detailed parameterizations schemes and major parameters of CICE are further covered in Appendix A. Two model resolutions of CICE are used: the nominally 1° grid of GX1V6 which is built-in in CESM, and the nominally 0.15° grid of TS015 [previously implemented in CESM (Xu et al., 2021)]. Notably, the TS015 grid has the horizontal dimension of 2400×1680 globally, and the mean grid resolution is about $7km$ in the Arctic region. Prominent, multi-fractal sea ice deformations can be simulated at this resolution (Xu et al., 2021).

For all the experiments, CICE is coupled to the Slab Ocean Model (SOM) and forced by the CORE-II dataset through the coupling framework in CESM. The CORE-II dataset is based on NCEP atmospheric reanalysis and further used in the Ocean

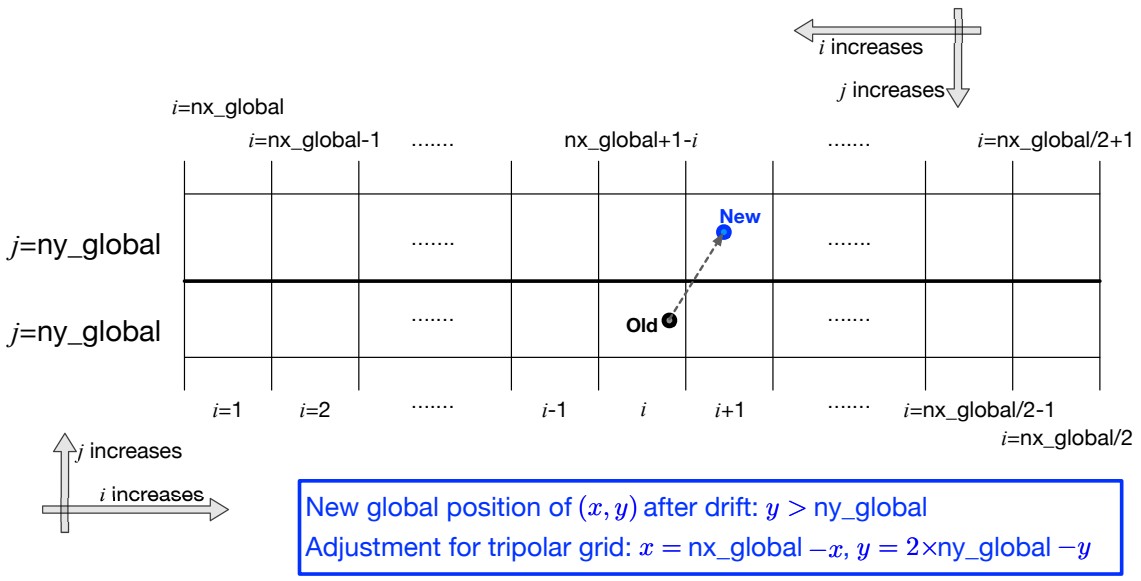

**Figure 4.** Support of Lagrangian point migration for tripolar grids. The model's global grid is of size $nx\_global$ by $ny\_global$, with the northern tripolar boundary shown. U-fold tripolar grids are supported (i.e., sea ice velocities are defined on the corner points of the Arakawa-B grid and along the northern boundary).

Model Intercomparison Projects (Griffies et al., 2016). It contains two separate forcing datasets: the Normal Year Forcing (NYF) which is annually repeating, and the Inter-Annual Forcing (IAF) which is for the years between 1948 and 2007. While the NYF dataset is usually used for the long-term simulations and the spin up of the ocean/sea-ice coupled model, the IAF dataset can be used for the hindcast of the historical states of the ocean and the sea ice (Wang et al., 2016). Both the NYF and the IAF datasets are used for the numerical experiments, covered in the following of the section.

The simulated Arctic sea ice climatology under NYF is shown in Fig. 5. The seasonality of the sea ice extent at both 1° and 0.15° (not shown) is consistent with the observational climatology (NSIDC). The overestimation of SIE during winter months mainly manifests in the peripheral seas of lower latitudes. In particular, in the Altantic Arctic region, the overestimation of SIE may be due to the absence of ocean heat transport of the SOM model. During summer, the SIE agrees well with the observation. In terms of the sea ice volume, there is general underestimation compared with PIOMAS (Schweiger et al., 2011). Thick, multi-year sea ice manifests north to the Greenland and the Canadian Arctic Archipelago (CAA), with the clockwise circulation in the Canadian Basin as controlled by the Beaufort High. With the generally good agreement of the modeled sea ice states to the observational datasets, we consider it sufficient for further analysis of the Lagrangian tracking. Moreover, further improvements of the model's simulation results are planned for future work, including the coupling with a dynamic ocean model as well as the tuning of parameterization schemes in CICE.





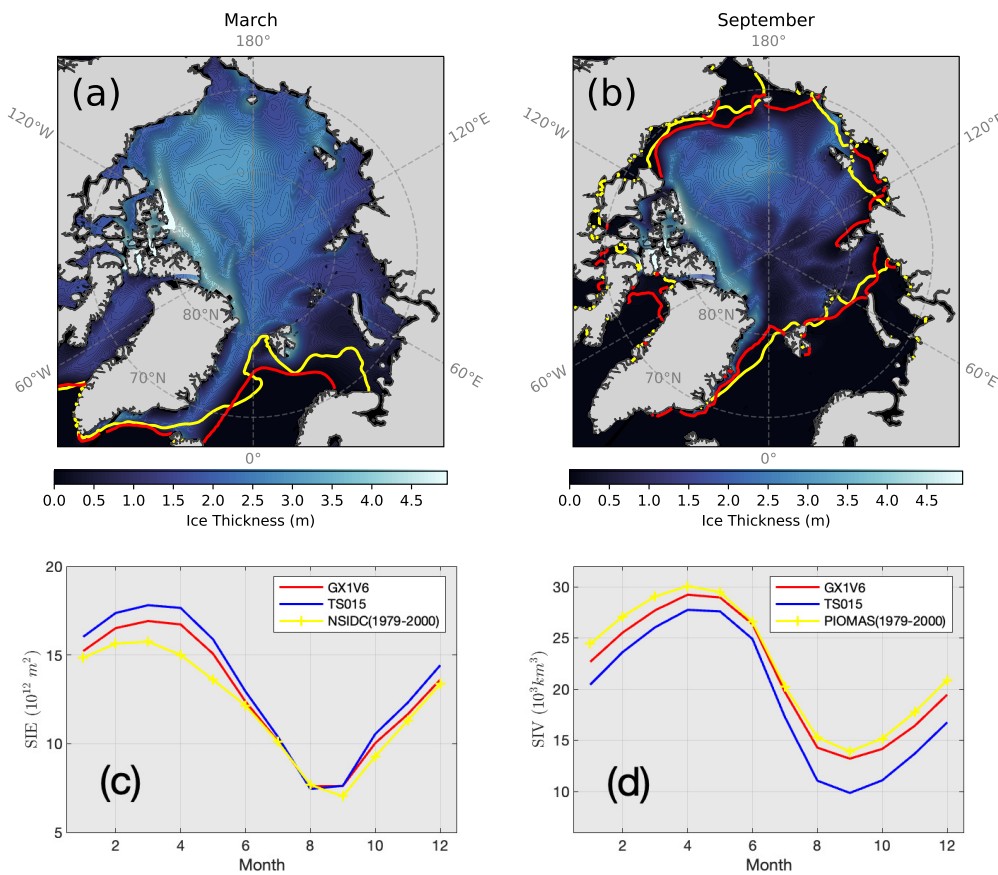

**Figure 5.** March (a) and September (b) sea ice thickness simulated by CESM and GX1V6 grid. The last 10-year's results are used to compute the thickness fields (filled contour) and the ice edge with sea ice concentration (SIC) at 15% (red line). The climatological sea ice concentration from NSIDC (based on passive microwave sensors) of the corresponding months are computed for the years between 1979 and 2000, with the sea ice edge shown by the yellow line (SIC=15%). The seasonal cycle of the Arctic sea ice extent (SIE) and the volume (SIV) are shown in panel (c) and (d), respectively. The simulation results from both GX1V6 and TS015 are included. For comparison, the SIE is based on NSIDC dataset, and the SIV is computed from the PIOMAS dataset for the same period (i.e., 1979 to 2000).



### 3.1 Climatology of Lagrangian points under NYF

Under the annually-repeating NYF dataset, we carry out the basic test of the Lagrangian tracking with the low-resolution, $1°$ grid. After the model has reached quasi-equilibrium after 20-year's spin up, we deploy one Lagrangian points at the center of every grid cell with sea ice ($SIC > 5\%$) in the Northern Hemisphere. The initial locations of the points are shown in Figure 6.a. With the summer melt and the transport to lower latitudes, the count of active Lagrangian points decreases with each year. After a full year since the deployment, the Lagrangian points outside the Arctic Basin are all lost due to melting (Fig. 6.b). The overall distribution of remaining points indicates the basin-scale drift of sea ice: (1) the anti-clockwise drift in the Beaufort Gyre; (2) the convergence of points to the north of Greenland/CAA; and (3) the transpolar drift from the eastern part of the basin and the outflow in the Fram Strait. After 5 years and 10 years (Fig. 6.c and d), the Lagrangian points are generally lost. The surviving points are mainly retained due to the accumulation in the Beaufort Gyre. Among all of the 3599 points that are originally in the basin ($80°N$ and north), about 2063 (57.3%) points are lost through melting in the basin within the first 20 years, while 1083 (30.1%) due to export from the Fram Strait (FS). In later years, the surviving points are mainly lost due to the circulation to the transpolar drift (hence the FS export) or to the eastern part of the basin with more prominent summer melt. The daily locations of the Lagrangian points are shown by the animation in the supplementary to the manuscript. It is worth to note that the NYF forcing dataset does not support direct comparison with observations. Hence the experiment results, including the simulated circulation of the Lagrangian points and the loss due to melt process, provides us with basic test of the Lagrangian tracking functionalities.

### 3.2 Validation of Lagrangian tracking under IAF

We further carry out the evaluate the Lagrangian tracking through the IAF-based simulation and the comparison with the buoy measurements. Based on the $1°$ grid (GX1V6) and the model's spun-up status under NYF, we carry out the historical simulation for the years between 1979 and 2001. For every day during the winter (December to March) at 0:00 (UTC), we deploy a Lagrangian point at every grid cell with the presence of sea ice. For each Lagrangian point, we track its location of up to 90 days. The daily instantaneous locations of the points are recorded and further compared with the observations.

The observational dataset from the International Arctic Buoy Programme (Rigor et al., 2008, IABP) during the same years is used for validation (data available at: https://iabp.apl.uw.edu/data.html, last access: 10 May 2023). Since we deploy Lagrangian points to the center of the grid cells, we use the following criteria to match to the physical buoys' locations of IABP. In total, 621 buoys with 49,004 hourly locations are available for comparison. We further split each buoy's continuous track into sub-tracks of 14 days for further comparisons. For each buoy's (sub-)track, we screen over the newly deployed Lagrangian points on its starting date. The Lagrangian point nearest to the track's starting location is located, and the simulated track is then compared against the observation. Leap years are ignored in the IAF dataset and the simulations, hence the buoys' tracks starting on February 29th are matched with the Lagrangian points on March 1st.

Figure 7 shows the matching between the buoys' and the corresponding Lagrangian points' locations during the winter of 1993–1994 (December to March). All of the sub-tracks, each covering 14-day's buoy's track, are shown. The two major sea



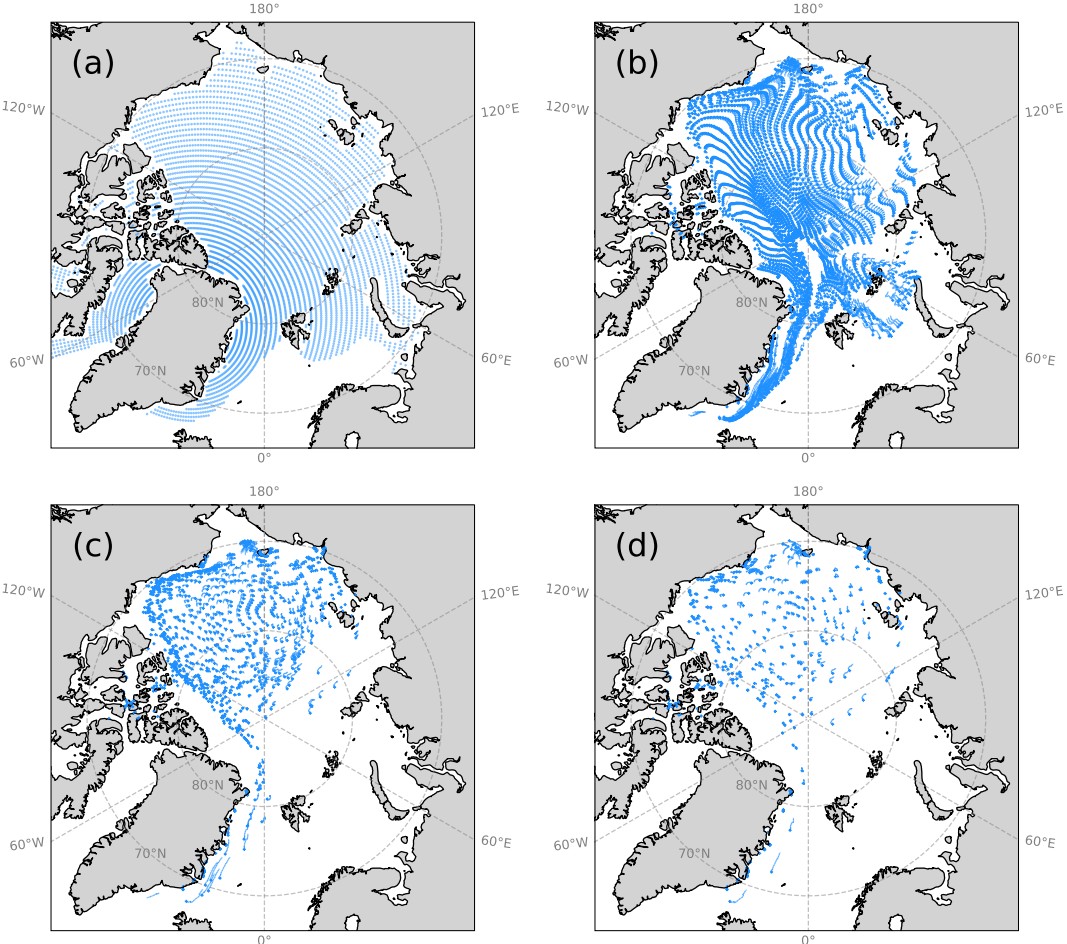

**Figure 6.** Lagrangian points (dots) at the initial deployment on Jan-1st (a), after 1 year (b), after 5 years (c) and 10 years (d). The previous locations of each point is also shown of up to 2 weeks (lines).

ice drift regimes include the clockwise circulation in the Canadian basin and the clockwise drift and export of the sea ice in the region east to $60°W$ and west to $130°E$. In both regimes, the modeled Lagrangian points' tracks are highly consistent with the observations.

Furthermore, Figure 8 shows for all the sub-tracks the distance between the locations of the buoys and the corresponding Lagrangian points. In total there are 2849 sub-tracks for comparison, each corresponding a buoy's drift within 14 days. For the

starting locations of the buoys' sub-tracks, the average distances to the matching virtual buoys are all within $50km$. The average grid size within the Arctic Basin is about $50km$, with finer (coarser) resolution in the area near (far from) the Greenland. For over 75% of the matching buoy pairs, the initial distance is less than $27km$. After 14-day's tracking, the distance gradually grows, but for 75% of all pairs, the distance is within $60km$. Besides, the median (mean) distance is $38km$ ($43km$). On average,



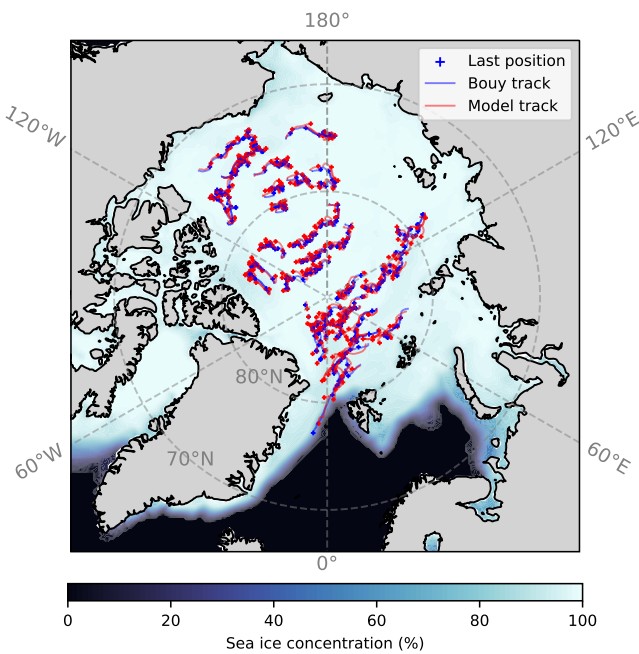

**Figure 7.** Simulated Lagrangian points (blue) and those of the corresponding IABP buoys (red) during the winter of 1993–1994. The last location (+) of each record and the track (line) of up to 14 days are shown for each buoy/Lagrangian point. The wintertime mean (DJFM) sea ice concentration is also shown in the background.

the average drift distance of buoys is $87km$ at 14-day scale. For fast moving buoys such as those entering the Fram Strait, the
distances between the Lagrangian points and the matching physical buoys are larger, but the relative error always remains low within the two-week's tracking time. In general, based on the Lagrangian tracking in CICE, the sea ice drift as simulated by the model matches well with buoy's observations. The tracking uncertainties may arise from the limited spatial resolution of the model, the uncertainty of the atmospheric forcing dataset, as well as the sea ice model's dynamics in simulating the observed drifts. Besides, due to the regular deployment of the Lagrangian points, there is no exact match of the buoys' initial locations.
Further attribution of the tracking error to the various contributing factors, including the model and the data's uncertainty, as well the initial location errors, is planned for future study.

### 3.3   Scaling analysis of sea ice kinematics in the $7km$-resolution simulation

We further carry out high-resolution simulations of sea ice kinematics and related analysis of the sea ice deformations. In particular, with the $7km$ resolution of the TS015 grid in the Arctic region, the model is capable to simulate the multi-fractal
sea ice deformations (Xu et al., 2021). The experiment is based on the NYF dataset and the model's quasi-equilibrium status after the spin-up process. The Lagrangian tracking is utilized for the diagnostics of the wintertime sea ice deformations and related statistics over multiple spatial and temporal scales. We first introduce the basic framework of scaling analysis and the



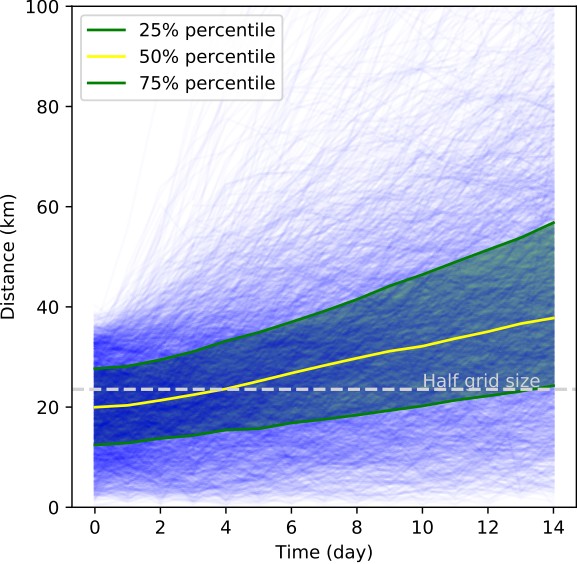

**Figure 8.** Distance between the modeled Lagrangian points and the corresponding buoys. The yellow line is the median, the lower and upper edges of the green shade are the 25th and the 75th percentiles, respectively. The horizontal bar (grey dashed line) marks half of the mean grid size ($\sqrt{dx \cdot dy}/2$) of the GX1V6 grid in the Arctic Basin.

use of Lagrangian tracking in Sec. 3.3.1. Furthermore, the detailed analysis of the spatial scaling (Sec. 3.3.2) and the temporal scaling and the spatial-temporal coupling (Sec. 3.3.3) are covered.

### 3.3.1 Lagrangian tracking and scaling analysis

The sea ice deformation rates ($\dot{\varepsilon}$'s) are defined with the simulated velocities. Since CICE is based on the locally orthogonal grid and Arakawa-B staggering, the model's velocity as well as the derived velocity of Lagrangian points are defined as fields of $u$'s and $v$'s. The grid invariant deformation rates, including the divergence rate ($\dot{\varepsilon}_{div}$) and the shear rate ($\dot{\varepsilon}_{shear}$) are then defined as follows. The total deformation rate ($\dot{\varepsilon}_{total}$) which is subjected for further scaling analysis, can be further defined with $\dot{\varepsilon}_{div}$ and $\dot{\varepsilon}_{shear}$.

$$\dot{\varepsilon}_{\text{div}} = u_x + v_y \tag{1}$$

$$\dot{\varepsilon}_{\text{shear}} = \sqrt{(u_x - v_y)^2 + (u_y + v_x)^2} \tag{2}$$

$$\dot{\varepsilon}_{\text{total}} = \sqrt{\dot{\varepsilon}_{\text{div}}^2 + \dot{\varepsilon}_{\text{shear}}^2} \tag{3}$$

In these equations, $u_x$, $u_y$, $v_x$ and $v_y$ are the spatial derivatives of $u$ and $v$ in the two orthogonal directions. The direct model output of the instantaneous and/or the temporal mean of the velocity fields, as well as the various deformation rates, are defined on the Eulerian model grid. In Figure 9 we show the maps of the daily-mean $\dot{\varepsilon}_{total}$ in the Arctic region for two typical





days of the experiment with TS015. LKFs manifest with the highly localized, anisotropic sea ice deformations. It is worth to note that the daily-mean velocity and the deformation fields (shown in Fig. 9) are inherent Eulerian, hence not proper for the scaling analysis. For example, the offline tracking (at hourly scale) should be carried out for further analysis [e.g., Bouchat et al. (2022)].


The velocity derivatives at various temporal/spatial scales (i.e., $\{u,v\}_{\{x,y\}}$'s) are further computed over the various Lagrangian patches. A set of adjacent Lagrangian points form an enclosed sea ice patch and start with the position on regular Eulerian grid points. Their locations change with the Lagrangian tracking, due to the sea ice drift and deformation. At certain time (delayed by $T$), their new locations are recorded and the displacement from their original locations are computed as $\Delta_x$'s and $\Delta_y$'s. Then the average velocity $\bar{u}$ and $\bar{v}$ for each Lagrangian point can be computed as $\bar{u} = \frac{\Delta_x}{\Delta_t}$ and $\bar{v} = \frac{\Delta_y}{\Delta_t}$ respectively. The spatial velocity derivatives of the Lagrangian patch are computed with the the patch's area ($A$) and the line integral of velocity over its outer boundary, following Kwok et al. (2008) and Rampal et al. (2019). Details of the computation are further covered in Appendix B.


Given the total deformation rates of all sea ice patches, we compute the average value of $\dot{\varepsilon}_{total}$'s within the similar spatial scale ($L = \sqrt{A}$). In particular, the $q$-th order of $\dot{\varepsilon}_{total}$'s, computed as $\dot{\varepsilon}_{total}^q$'s, along with their average value are also computed for $q \in \{0.5, 1, 1.5, 2, 2.5, 3\}$. By the scaling law (Marsan et al., 2004), under the spatial scale of $L$ and the temporal scale of $T$, we have:


$$
\langle \dot{\varepsilon}^q(T,L) \rangle \quad \sim \quad T^{-\alpha(q)} \tag{4}
$$

$$
\langle \dot{\varepsilon}^q(T,L) \rangle \quad \sim \quad L^{-\beta(q)} \tag{5}
$$

Where $\alpha(q)$ and $\beta(q)$ are the structure functions of the temporal and the spatial scaling. For a specific value of $q$, we carry out the linear fitting of $\langle \dot{\varepsilon}^q(T,L) \rangle$ with respect to $T$ (or $L$) to estimate the value of $\alpha$ (or $\beta$). For the observed multi-fractal deformations of the sea ice (Marsan et al., 2004), we witness convex structure functions for both $\alpha(q)$ and $\beta(q)$ with respect to $q$. Correspondingly, $\alpha(q)$ and $\beta(q)$ can be fitted as follows, with the fitted parameter of $a$ and $c$ both larger than 0.


$$
\beta(q) \quad = \quad a \cdot q^2 + b \cdot q \tag{6}
$$

$$
\alpha(q) \quad = \quad c \cdot q^2 + d \cdot q \tag{7}
$$

In order to evaluate the simulated sea ice deformations, we deploy a Lagrangian point at the center of each grid cell for every model day at the time of 0:00. The maximum allowed life span of each Lagrangian point is 30 days. As a consequence, the temporal scaling of up to 30-day can be studied (i.e., the maximum value of $T$ at 30-day). The model regularly reports the location of every Lagrangian point every 6 hours (i.e., the minimum value of $T$ at 6-hour). Both parameters can be configured at the compile time of the CICE model. The model grid's native resolution in the Arctic is about $7.3km$. Given that the effective resolution is coarser (Xu et al., 2021), we evaluate the deformations at the spatial scale from $30km$ to $480km$ (4 times to 64 times the grid's resolution). For the scaling analysis, we limit the initial locations of all Lagrangian points to be at least $400km$


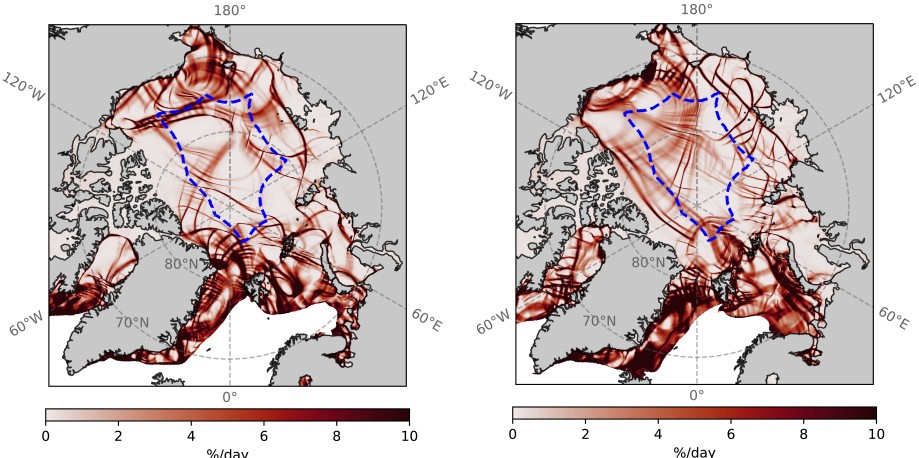

**Figure 9.** Daily-mean sea ice total deformation on Dec-20 (left) and Feb-6 (right) simulated by CESM and TS015. The region subjected to further scaling analysis is outlined by the dashed blue line in each panel.

away from land (i.e., within the outlined region in Fig. 9). The regions near the coast and the continental shelf break, where constant deformation features persist, are therefore excluded from the scaling analysis.

After the model's spin up, we output a whole winter's Lagrangian tracking results, together with the daily mean Eulerian fields of the sea ice velocity and the deformation. Furthermore, we also study the difference between the Lagrangian tracking-based scaling analysis and that based on the temporal/spatial mean fields of the model output on Eulerian locations.

### 3.3.2   Spatial scaling

We carry out spatial scaling analysis for two typical periods during the winter, focusing on 4 typical temporal scales: 1-
day, 3-day, 10-day, and 30-day. Figure 10 and 11 show the results around Dec. 20th and Feb. 6th, respectively. At $q = 1$ and 1-day scale, the spatial scaling of sea ice deformation are both very shallow: $\beta = 0.0623$ and 0.121, respectively. These two days correspond to different strengths and depths of the Beaufort High and slightly different Arctic Oscillation indices, and consequently the drift and deformation fields are distinct [see Fig. 6 and 11 of Xu et al. (2021)]. At different orders of momentum, the mean deformation rates all follow the power-law scaling. On higher orders ($q > 1$), large deformation events
are more dominant in the mean deformation rate, corresponding to steeper scaling (i.e., larger values of $\beta$). Furthermore, the value of $\beta$ grows nonlinearly with $q$. With the increase of the moment order, the deformation rate decreases faster with respect the increase of the spatial scale ($L$). Correspondingly, the convex shape of $\beta(q)$ indicates that the model simulates multi-fractal sea ice deformations.

For the period of study surrounding Dec. 20th, at larger temporal scales from 1-day to 30-day, the average deformation rates
decrease at all spatial scales and moments of order. More importantly, the value and the shape of $\beta(q)$ also change significantly with the temporal scale. At $q = 1$ and the 1-day temporal scale, the value of $\beta$ is 0.0623. For larger temporal scales, the mean





deformation rate gradually decreases, which also applies to the structure function of $\beta$. In particular, the convexity of $\beta(q)$, indicated by the fitted value of $a$, decreases drastically from 0.137 (1-day) to 0.064 (30-day). The change in $a$ indicates that there is strong coupling between the spatial and temporal domain for the sea ice deformation. Similar results hold for the

period around Feb. 6th, with the two following major differences: (1) more convex $\beta(q)$ around Feb. 6th than Dec. 20th; (2) the faster decrease of $a$ with respect to $T$ (i.e., the temporal scale). The most significant drop in $a$ occurred from the 3-day to the 10-day scale for Feb. 6th (from 0.151 to 0.101). For comparison, that for Dec. 20th is between 10-day to 30-day (from 0.093 to 0.064). Regarding the differences between the two periods, we conjecture that the process-dependent deformations are the major cause, including the strength of the spatial-temporal coupling. Further analysis is needed for the attribution of

these differences to various factors, including the sea ice status, its deformation history, as well as the forcings.

  For comparison, we also shows the results based on Eulerian deformation fields in both Figure 10 and 11, including the power-law fittings and the structure functions of $\beta(q)$. The method details are introduced in Appendix B. For both periods of study, the scaling analysis with Lagrangian tracking results show steeper $\beta$ functions with higher convexity across all temporal scales (i.e., larger values of $a$'s). For Feb. 6th, the difference between the two is more pronounced. Note that the scaling

analysis should be carried out based on the Lagrangian perspective. The objective for the comparisons is to demonstrate that there exist systematic differences of the scaling analysis when using the Eulerian model outputs. The quantitative differences of the deformation rates may arise from: (1) that the deformation events are misaligned between the Eulerian and the Lagrangian perspective, and (2) the secondary effect of changing shapes and scales when the sea ice deforms within the Lagrangian perspective, which is not captured by the Eulerian framework.

### 3.3.3 Temporal scaling and spatial-temporal coupling

For the temporal scaling analysis, we focus on the spatial scale of the model's effective resolution ($L = 22km$). Figure 12 shows the results for the outlined region in Figure 9 in the month of December (left) and February (right). The analyzed temporal scale is in the range between 1-day and 30-day. Similar to the spatial scaling results, the power-law scaling is witnessed for the temporal scaling (top panels of Fig. 12). Also, convex structure functions of $\alpha(q)$ are present for both months, with the fitted

value of $c$ as 0.063 and 0.059, respectively. This results indicate that the model also simulates multi-fractal sea ice deformation in the temporal domain.

  Evidently, the power-law scaling at sub-daily scales is much shallower than that between 1-day and 30-day. Similar behavior of the sea ice is observed in Oikkonen et al. (2017), which covers the temporal scale from about 10 minutes. We consider the shallower scaling in the simulations is qualitatively consistent with the observation. However, although the model's dynamics

timestep is sufficiently short (see Appendix A), the temporal resolution of the forcing dataset is much coarser, at 6-hourly. In order to fully study the simulation of the sub-daily sea ice deformations, we need high-frequency forcing datasets or the coupled simulation with the high-resolution interactive atmospheric component (Zhang et al., 2023).

  Similar to the analysis in Section 3.3.2, we also compute the temporal mean Eulerian deformation fields and the equivalent scaling analysis. Since we only output daily model fields, the analysis is limited to the temporal scale between 1-day and

30-day (middle row of Fig. 12). Apparent power-law scaling is witnessed for both months. However, the structure function of

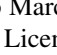



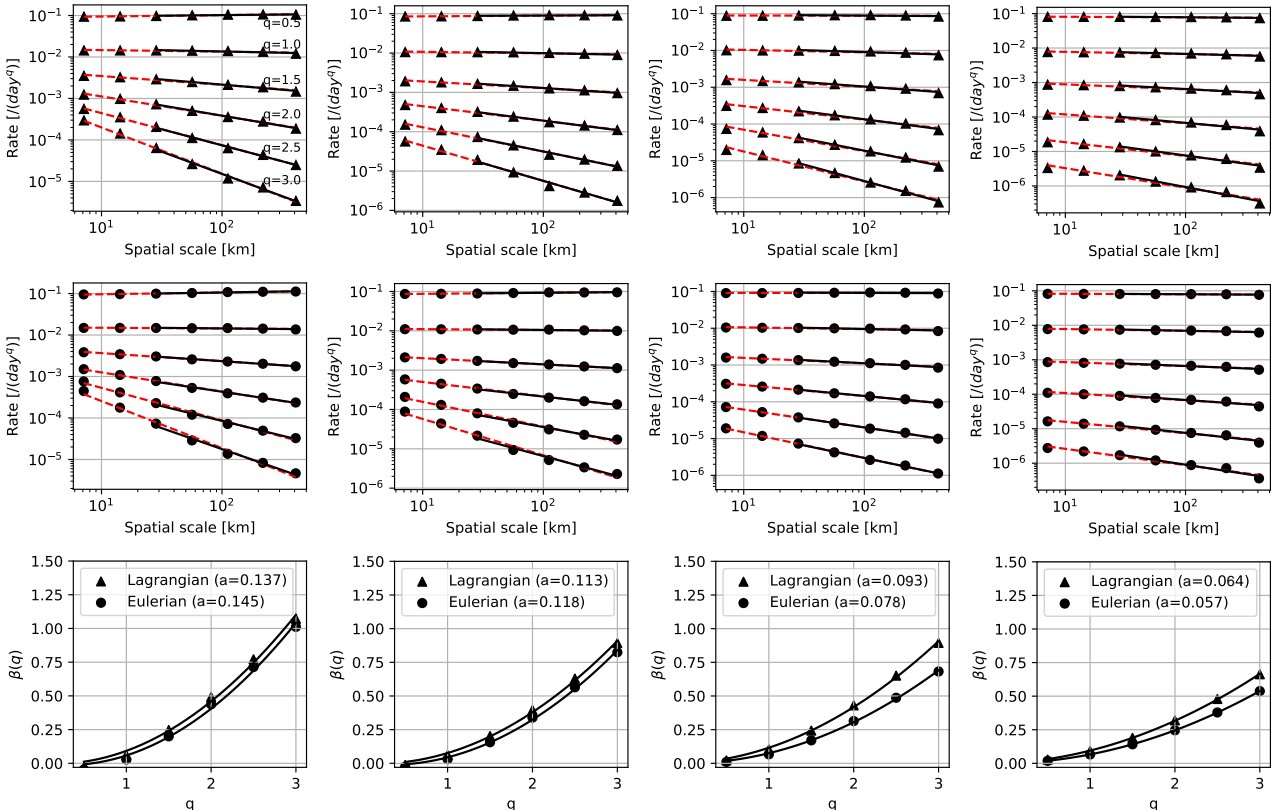

**Figure 10.** Spatial scaling around Dec. 20th. Four temporal scale are evaluated: 1-day (first column), 3-day (second column), 10-day (third column), and 30-day (last column). Results with Lagrangian tracking (triangles) are compared with that based on temporally mean Eulerian deformation fields (dots). Black lines in each panel represent the fittings with the spatial scale of $30km$ and coarser, and red dashed lines the fittings with the whole resolution range of TS015 ($7.3km$ and coarser). The fitted parameter of $a$ for the structure function $\beta(q)$ is shown in the legend.

$\alpha(q)$ with the Lagrangian tracking is systematically less convex than that based on Eulerian fields, with lower values of $c$ for both months (bottom row of Fig. 12). This result, together with that in Section 3.3.2, show that the scaling analysis should be carried out in the Lagrangian framework. Although the scaling analysis with the model's Eulerian mean fields yields apparent multi-fractal sea ice deformations, the results are systematically different from the those based on the Lagrangian analysis.

Furthermore, we conduct the initial analysis of the spatial-temporal coupling during the whole winter (December to March). For the region of study (outlined in Fig. 9), we utilize all of the Lagrangian tracking results by forming and tracking Lagrangian patches at various spatial and temporal scales. Specifically, for each combination of the spatial and the temporal scale (i.e., $L$ and $T$), we form Lagrangian patches that satisfy the following criteria: (1) they start at interleaved Eulerian grid locations separated by $L/2$ in both directions, and (2) the time difference between their starting time and 0:00 of Dec. 1st separated

by $n \cdot T/2$, where $n \in \{0, 1, 2, 3, ...\}$. With these two criteria, we in effect split the Lagrangian tracking results into sea ice



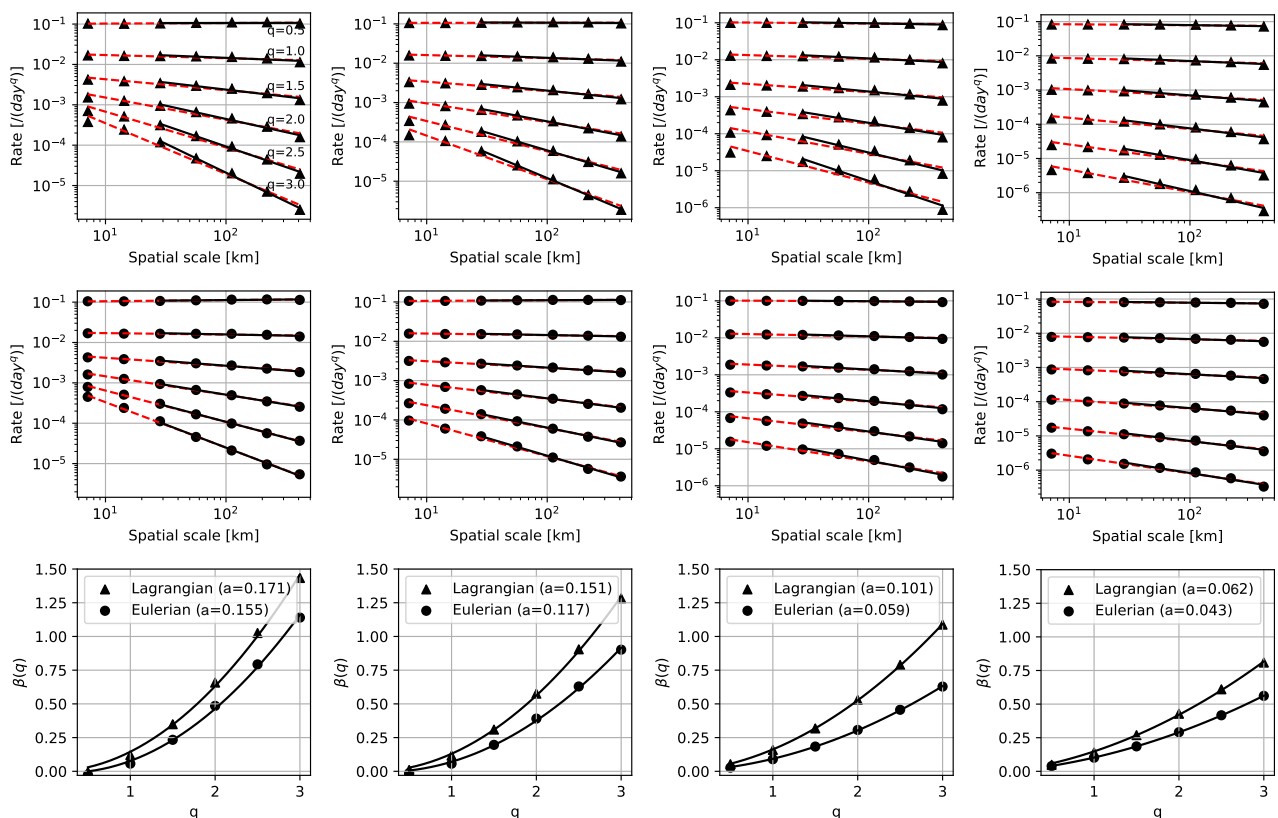

**Figure 11.** Same as Fig. 10, but for Feb. 6th.

Lagrangian patches with 50% temporal and spatial overlapping. As a result, we not only attain full coverage of the study area/period, but also avoid potential sampling issues associated with the changing weather regimes.

The relationship between the curvature parameter of the spatial (temporal) scaling structure function and the temporal (spatial) scale is shown in Figure 13.a (b). There is evident coupling between the spatial domain and the temporal domain, with
decreased curvature of the spatial (temporal) structure function at larger temporal (spatial) scales. In particular, for the temporal scaling, there is a good fit of the curvature parameter $c$ to the Power-Law (Fig. 13.b), which is consistent with various estimations based on in-situ and satellite-based remote sensing observations (Rampal et al., 2008; Marsan and Weiss, 2010). However, for the spatial scaling, the relationship of $a$ to the temporal scale is much flatter and in particular, less convex for a Power-Law fit (Fig. 13.a). We note that during different periods of the winter, the sea ice drift and deformation patterns are
highly heterogeneous (due to changing weathers), with the sea ice condition undergoing significant changes. For example, the spatial scaling exponent shows large temporal variability (Rampal et al., 2019) and is likely linked to the atmospheric forcing patterns (Xu et al., 2021). Therefore, we adopt another fitting with the $log$-quadratic form: $y = p \cdot log(x)^2 + q$. This new form yields much higher fitting to the relationship between the curvature parameter $a$ and the temporal scale $T$ ($R^2$ from 0.882 to



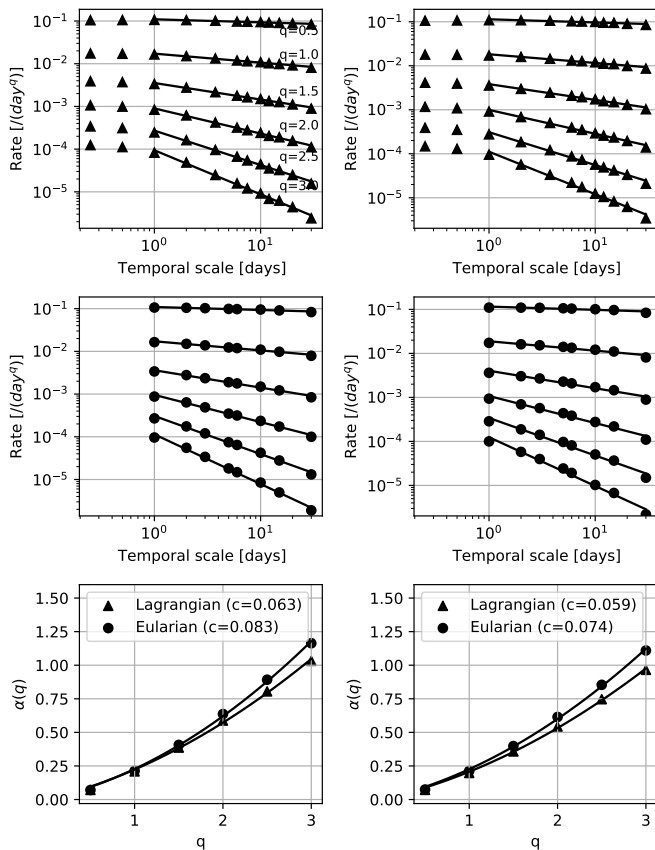

**Figure 12.** Temporal scaling for December (left) and February (right) at the spatial scale of $22km$ (i.e., $3\times$ the original grid resolution). Similar to Fig. 10 and Fig. 11, the results with Lagrangian tracking and Eulerian mean deformation fields are shown by triangles and dots, respectively. Note that the fitting are only computed for 1-day or coarser temporal scales (indicated by the abscissa range of the fitted lines).

0.989). The efficacy of the Power-Law fitting, as well as the full analysis of the spatial-temporal coupling as simulated by CICE
is beyond the scope of this study. Especially, the sensitivity to sea ice rheology model and other dynamic processes, is planned
for both CICE standalone and coupled experiments with CESM.

## 4    Summary and discussion

We design and implement the online Lagragian tracking in the sea ice model of CICE. It is incorporated in the dynamics
process of CICE, and fully supports the domain decomposition and parallel computing in CICE. Compared with the common
practice of offline sea-ice tracking (Bouchat et al., 2022; Sumata et al., 2023), the online tracking has several advantages. First,
the offline tracking usually requires the storage of high-frequency sea ice drift data, which is not needed for the online tracking.
Second, since the online tracking is carried out per dynamics step, the tracking error is minimal. For offline tracking, the

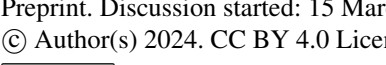



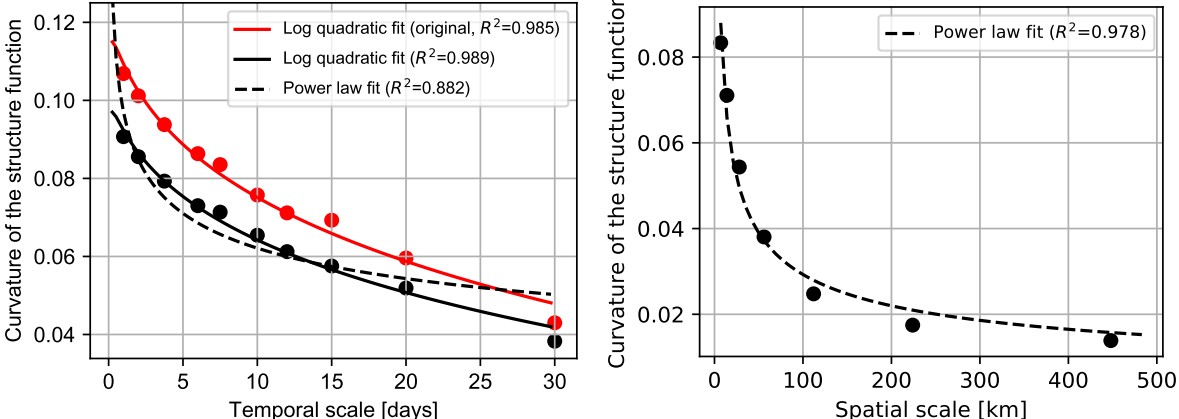

**Figure 13.** Temporal-spatial coupling (Dec.-Mar.) analysis of Lagrangian tracking deformation fields: The curvature of the structure function of the spatial(temporal) scaling exponent as a function of temporal(spatial) scale is shown in panel a(b). The solid and dashed lines represent the logarithm-quadratic fitting of $y = p \cdot (log(x))^2 + q$ and the power-law fitting of $y = p \cdot x^q$, respectively. The red (black) line in the first panel shows the fitting with all the spatial scales (the scales with the model's effective resolutions). The $R^2$ of each fitting in top panels are shown, respectively. Scaling analysis with both time scale and spatial scale as independent variables are shown in (c), correspondent structure functions are shown in (d).

tracking uncertainty arises from the relative coarse temporal mean (e.g., daily) sea ice drift fields, as well as the variable tracking frequency [see examples in Bouchat et al. (2022)]. Moreover, the study of sea ice dynamics at small temporal scales (such as
sub-daily motions) with offline tracking requires even finer resolution of the stored sea ice drift. The study of these processes is readily supported by the online tracking [see Fig. 12 and also Oikkonen et al. (2017)]. Computationally, the Lagrangian tracking only induces a small overhead to the overall simulation. For example, for the daily deployment of Lagrangian points in the $7km$ experiment in Section 3, there are on average 30 to 35 million active Lagrangian points, and the extra computational overhead is less than 3%. Therefore, the current implementation is capable to support high-resolution simulations with a large
quantity of Lagrangian points.

In Section 3.3 we carried out the spatial and the temporal scaling analysis with the Lagrangian tracking based on high-resolution simulation during winter. As shown, the CICE model simulates multi-fractal sea ice deformations both spatially and temporally, as well as the tight coupling between the spatial and the temporal domain. The scaling properties of the simulations are consistent with observed statistics based on high-resolution synthetic aperture radar satellite payloads. In particular, we
compare the Lagrangian-based scaling statistics with the counterparts based on Eulerian model outputs. Results highlight the importance of using Lagrangian-based diagnosis for the scaling analysis: although the analysis based on Eulerian output also yields convex structure functions, the fitted convexity parameters are significantly different from those based on the Lagrangian perspective.



In order to compare with the observed sea ice drift and deformation (e.g., RGPS) as well as the scaling statistics, we
plan to carry out high-resolution, atmospherically-forced historical simulations with the CICE/CESM model assisted with
the Lagrangian tracking. Consequently, fine tuning of the sea ice model parameters are needed for: (1) the optimization of
the simulation of various sea ice parameters such as sea ice thickness, and (2) improved comparability to the observed sea
ice deformations. Especially, in Bouchat and Tremblay (2020) the authors proposed the new estimation of deformation rates
regarding their uncertainties. Better consistency with model simulations is attained with further incorporation of the model
simulation and tracking uncertainties (Bouchat et al., 2022). We plan to incorporate the method and carry out systematic
evaluation of the model's capability to simulated the observed multi-scale sea ice deformations. In particular, the sensitivity to
the sea ice rheology is a key aspect of the planned future work (Tsamados et al., 2013; Bouchat et al., 2022).

In this first version of the Lagrangian tracking in CICE, virtual buoys are supported, which can be deployed at regular
time intervals and on regular locations. Section 3.2 shows that with the pairing between the physical buoys and the respective
virtual buoys, the simulated tracks are generally consistent with observations. For future development, we plan to support
the deployment of physical buoys in the simulation at prescribed time and locations, so that we can better compare with
observational datasets such as IABP. Moreover, key Lagrangian tracking related parameters (in Appendix B) is planned to be
configured through a subset of the namelist in the future development plan. Finally, the current design of Lagrangian tracking
is based on the Arakawa-B grid sea ice dynamics and the transport remapping scheme. For the compatibility with ongoing
development in CICE6 (https://github.com/CICE-Consortium/CICE/releases), we also plan to incorporate flux-form based
Lagrangian tracking which is suitable for Arakawa-C grid sea ice dynamics (Lemieux et al., 2024).

The Lagrangian tracking can be used in a variety of science questions and applications related to sea ice. Large-scale
Lagrangian survey can be carried out in the simulations, complementary to buoys' measurements which are limited in terms of
spatial and temporal coverage. For example, virtual buoys can be deployed in historical simulations, which enables systematic
study of the thermodynamic and dynamic history of the sea ice (Sumata et al., 2023). Furthermore, CICE is widely adopted
for the high-resolution sea ice operational systems (Smith et al., 2016; Yang et al., 2020). The online tracking of sea ice in key
regions can be carried out for sea ice forecasts, in order to support operations and ship navigation in polar waters.

*Acknowledgements.* The authors would like to thank the editors and referees for their invaluable efforts in improving the manuscript.
This work is mainly supported by the joint project of INTERAAC co-funded by the National Key R&D Program of China (grant no.:
2022YFE0106700) and the Research Council of Norway (grant no.: 328957). JL is supported by the National Key R&D Program of China
(grant no.: 2018YFA0605900). SX is also partially supported by the National Natural Science Foundation of China (grant no.: 42030602),
the International Partnership Program of Chinese Academy of Sciences (grant no.: 183311KYSB20200015), and the Research Council of
Norway (grant no.: 328957). The authors would also like to acknowledge the computational and technical support from the National Super-
computing Center at Wuxi for the numerical experiments.





**Table A1.** Model parameters of CICE

| Parameter | Value | Notes |
|---|---|---|
| DT | $3600 s$ (GX1V6) and $1800 s$ (TS015) | Time step for sea ice dynamics |
| NDTE | 120 (GX1V6) and 960 (TS015) | EVP subcycling count |
| $C_f$ | 17 | Empirical parameter for frictional energy dissipation |
| $a^*$ | 0.5 | $e$-folding scale for participation function during ridging |
| $\mu_{rdg}$ | $4 \sqrt{m}$ | $e$-folding scale for ice ridging |
| $\rho_s$ | $330 \ kg/m^3$ | Snow density (used in D-E) |
| $R_{fresh}$ | $100 \ um$ | Freshly-fallen snow grain radius (used in D-E) |
| $R_{nonmelt}$ | $500 \ um$ | Seasoned snow grain radius (used in D-E) |
| $R_{melt}$ | $1000 \ um$ | Melting snow grain radius (used in D-E) |

*Code and data availability.* The original codebase of the CESM model (version 2, https://www.cesm.ucar.edu/models/cesm2) and the associated CICE model (version 5) is available through: https://github.com/ESCOMP/CESM/tree/release-cesm2.2.2 (last access: 2024-Jan-20). The sea ice concentration data is available at NSIDC through: https://noaadata.apps.nsidc.org/NOAA/G02135/ (last access: 2024-Feb-21). The sea ice thickness dataset of PIOMAS is downloaded from: https://psc.apl.uw.edu/research/projects/arctic-sea-ice-volume-anomaly/data/ (last access: 2024-Feb-21). The buoy tracks from IABP program are downloaded from: https://iabp.apl.uw.edu/Data_Products/BUOY_DATA/
3HOURLY_DATA/ (last access: 2024-Feb-21).

The Lagrangian tracking in CICE (version 5) and the sample Lagrangian tracking output are available at: https://zenodo.org/records/10791399 (last access: 2024-Mar-7).

The animation produced with the Lagrangian tracking with the $1°$ grid under NYF dataset is accessible as supplementary material of this article.

**Appendix A: Model configuration and parameters of CICE**

The CICE (version 5.0) is configured with 5 discretized thickness categories for the Ice Thickness Distribution (ITD), 8 and 3 layers for sea ice and its snow cover, respectively. The Elasto-Viscous-Plastic (EVP) rheology model is used for all experiments. The ice strength parameterization follows that in Rothrock (1975), which relates the ice strength to the gain of the potential energy during the ridging process. The ridging/rafting parameterization uses an exponential distribution form of the ridged

ice in the ITD. For sea ice thermodynamics, the mushy layer physics parameterization is adopted (Turner et al., 2013). The Delta-Eddington (D-E) scheme is used for the radiation processes (Holland et al., 2012). Table A1 lists the key parameters of the model. The parameters are the same across all numerical experiments by default, except the dynamics time step and the EVP subcycling count (NDTE). These two parameters are set differently between the experiments with GX1V6 and those with TS015. The shorter time step is adopted for TS015 to ensure numerical stability, and the value of NDTE is further enlarged for

the numerical convergence of the EVP rheology (Lemieux et al., 2010; Xu et al., 2021).





## Appendix B: Scaling analysis with Lagrangian tracking and Eulerian fields

We carry out the scaling analysis of the sea ice deformations at various spatial and temporal scales (Rampal et al., 2019; Bouchat and Tremblay, 2020). The Lagrangian tracking results, as well as the model outputs on Eulerian grids, are used to compute the mean sea ice deformation rates at the corresponding scales. A set of Lagrangian points form an enclosed Lagrangian patch, for

which we compute the spatial derivatives of velocities (i.e., $u_x$, $u_y$, $v_x$ and $v_y$) as follows:

$$
\begin{aligned}
u_x &= \frac{1}{A} \oint u \, \mathrm{d}y \\
u_y &= -\frac{1}{A} \oint u \, \mathrm{d}x \\
v_x &= \frac{1}{A} \oint v \, \mathrm{d}y \\
v_y &= -\frac{1}{A} \oint v \, \mathrm{d}x
\end{aligned}
\tag{B1}
$$

These derivatives are used to compute the deformation rates as in Eqs. 2 to 3. The line integral in Eqs. B1 is carried out over the Lagrangian patch that originally covers a regular rectangular domain. Then each point on the patch's outer boundary, called a vertex, has its location of $(x_i^j, y_i^j)$ at the time of $t_j$, and the new location of $(x_i^{j+1}, y_i^{j+1})$ at the time of $t_{j+1}$. Figure B1

shows the schematics for a Lagrangian patch with 4 corner points. We can estimate its mean velocity $(u_i, v_i)$ as the ratios of the displacements to the time difference: $(x_i^{j+1} - x_i^j)/(t_{j+1} - t_j)$ and $(y_i^{j+1} - y_i^j)/(t_{j+1} - t_j)$. Furthermore, in order to compute the line integrals at time $t$, we compute the mean velocity on each edge as follows (for the example in Fig. B1):

$$
u_x = \frac{1}{A} \sum_{i=1}^{4} \frac{1}{2} (u_{i+1} + u_i)(y_{i+1} - y_i),
\tag{B2}
$$

where $A$ is the area the set of Lagrangian points cover:

$$
A = \frac{1}{2} \sum_{i=1}^{4} (x_i y_{i+1} - x_{i+1} y_i),
\tag{B3}
$$

In order to complete the line integral, we set $x_5 = x_1$, and similarly the corresponding values of $y_5$, $u_5$, and $v_5$. For larger spatial scales, we start with a larger set of Lagrangian points that have their original locations on the Eulerian grid locations and cover a locally rectangular area. Then the velocities, their spatial derivatives, and the line integrals are computed, followed by the computation of the deformation rates. This is equivalent to the aggregation of all the smallest rectangular units of 4

Lagrangian points that constitute the whole rectangular area. Note that at the beginning of each model day, we deploy the Lagrangian points on all Eulerian grid locations, so that we attain full spatial and temporal coverage, from the daily scale and above. Besides, we further exclude the cases with the size change by the factor of 2 or larger.

For comparison, we also compute the equivalent deformation rates based on Eulerian fields. We directly obtain the temporally mean velocities on each Eulerian grid location, and compute the velocity gradients correspondingly (Fig. B1.b). Different





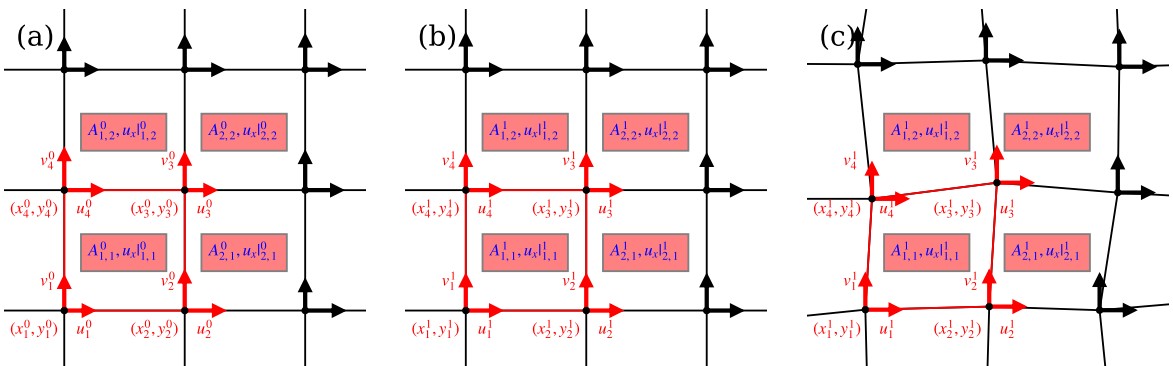

**Figure B1.** Schematics of the scaling analysis with Lagrangian tracking and the equivalent Eulerian model outputs. The locations of Eulerian and Lagrangian points at the initial time $t_0$ are shown in panel (a), and those at time $t_1$ are shown in panel (b) and (c), respectively. The spatial scale $L$ is defined as the square root $A$ which is the area of the region covered by the four Lagrangian/Eulerian points. Note that in our analysis, the initial locations of the Lagrangian points are the regular, Eulerian grid points. For comparison, on the same time scale (i.e., $T = t_1 - t_0$), we compute the model-output mean velocities on the Eulerian grid points, as well as the displacement-based velocity estimations for Lagrangian points.

spatial scales correspond to the various areas that corresponds to different rectangular grid cells, which do not change througout the simulation.

For both Lagrangian statistics and the Eulerian counterparts, we only compute the deformation field that is contained by the areas covered at the coarsest spatial scale. This practice ensures that there is no preferential sampling of the deformation events.

A wide range of both spatial and temporal scales are adopted for the computation of the deformation rates. The statistics of $\dot{\varepsilon}(T, L)$, which is the deformation rate at the spatial scale of $L$ and the temporal scale of $T$, include its $q$-th order [i.e., $\dot{\varepsilon}^q(T, L)$] and the mean value are computed over the studied area. It is worth to note that the spatial scale of $L$ varies within the studied domain, as well as with time for the Lagrangian tracking. Therefore, both the mean value of $\dot{\varepsilon}^q(T, L)$'s and that of the spatial scales of $L$'s are used for the scaling analysis.

**Appendix C: Major parameters of Lagrangian tracking**

Tab. C1 shows the major user-specified parameters for running the Lagrangian tracking in CICE. In the current implementation, they are configured at the compile-time of the model. In the future, we plan to implement certain parameters to be configurable through the namelist.





**Table C1.** User-prescribed parameters of Lagrangian tracking

| Parameter | Type | Default value | Description |
|---|---|---|---|
| `LAGR_BUFFER_SIZE_PARAM` | Integer | 100000 | Size of the pool of Lagrangian points |
| `LAGR_BNDY_SIZE_PARAM` | Integer | 400 | Per-cell maximum count of migrating Lagrangian points |
| `lagr_aice_thres` | Real | 0.05 | Threshold value of SIC for detecting melt events |
| `LAGR_REPORT_INTERVAL` | Real | 18000 $s$ | The interval of reporting Lagrangian points' status |
| `LAGR_ACTIVATION_INTERVAL` | Real | 86400 $s$ | Virtual buoy activation interval (in seconds) |
| `LAGR_ACTIVATION_DNSTY` | Real | 1.0 | Deployment density of virtual buoys (1 per cell) |
| `LAGR_VIRTUAL_BUOY_MAX_LIFE_DURATION` | Real | 2592000 $s$ | Maximum lifetime of virtual buoys (in seconds) |

*Author contributions.* JL and SX conceived the work. SX carried out the design and the implementation of the Lagrangian tracking in CICE.
SX, CN and ZY carried out the numerical experiments. NC and SX analyzed the results. All the authors contributed to the writing of the
manuscript.

*Competing interests.* The authors declare that they have no conflict of interest.



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
