# Peer review of "Lagrangian tracking of sea ice in Community Ice CodE (CICE; version 5)"

_Geoscientific Model Development, 2024_

## Author Comment (AC1)

The authors would like to thank the editor and the referee's comments on our manuscript. Following the comments, we make the following replies and corresponding revisions to the manuscript. Each item of the original comments from the referee is in *blue italic*, followed by our reply. Moreover, in the marked version of the revised manuscript, the revisions are highlighted with 'REV1'.

**REVIEW 1**

*This study introduces an online Lagrangian tracking module implemented in the CICE under the coupled model system of CESM to enhance Lagrangian diagnostics in sea ice models. The authors validated their module through numerical experiments focusing on sea ice deformations and kinematics. These experiments revealed multi-fractal characteristics in both spatial and temporal domains, as well as spatial-temporal coupling. The novelty of this work lies in the development of the Lagrangian tracking module and its emphasis on the importance of the Lagrangian perspective. This contributes significantly to the field of sea ice research. While the work is of sufficient quality and depth for publication, I have a few minor inquiries that I would like to discuss:*

*1. The authors outline the general structure of Lagrangian tracking within the time step of CICE. Lagrangian tracking was conducted prior to ridging and rafting. However, in these phenomena, multiple Lagrangian points may overlap or intersect. It would be helpful to clarify whether, in cases of strong ridging or rafting, these overlapping or intersecting Lagrangian points were considered as a single point or if they were treated separately in the tracking process. Could you elaborate more about ridging and rafting?*

**Reply**: we totally agree with the referee on this comment on sea ice ridging. During ridging events, especially over longer periods during which several ridging events are possible, the overlapping between Lagrangian points' tracks is possible. In the Lagrangian tracking we implement in CICE, these points are considered individually, fully allowing these cases.

We would also like to emphasize that: the linear kinematic features (LKFs), such as ridging/shear belts, mostly manifest at the spatial scale larger than the model grid's native resolution. The potential reason is that the effective resolution of the model is coarser for resolving sea ice dynamic processes. Consequently, we also carry out the scaling analysis for the spatial scales beyond 3x the original resolution of the grid.

On the other hand, at each step, the Lagrangian tracking is carried out within a single grid cell or between adjacent grid cells (i.e., on the scale of the grid cells). This limitation arises naturally with the numerical stability limitation in CICE. Therefore, there exists a scale separation between Lagrangian points' tracking and the detectable LKFs.

A further clarification is: during ridging/rafting, the shrinkage of the cell area occurs, but the Lagrangian points in the cell do not *feel* the ridging directly.

Lastly, the potential loss of Lagrangian points due to sea ice mechanical processes is planned for future development, but not available in the current implementation.

2. *The authors discuss the climatology of Lagrangian points under NYF (figure 6), supplemented with video. This discussion is valuable for understanding the functionality of Largrangian tracking module and visualizing sea ice export and melting. While direct comparison to observations is constrained, it would be beneficial to discuss the convergence of results concerning the number of Lagrangian points. In addition, exploring sea ice transport using non-uniform or localized distributions of Lagrangian points could provide further insights into sea ice dynamics. Understanding how variations in Lagrangian point distribution influence sea ice transport patterns would enhance the comprehensiveness of the study's findings.*

**Reply**: as pointed out by the referee, the NYF is annually repeating, therefore only representative of the climatology of the sea ice status, and not directly comparable to observations. Hence we further provide a newly added figure (below) to demonstrate the effect of AO's different phases on the points' track, using the NYF-based experiment.

[Figure]

Figure. Sea-level pressure (SLP) and the tracks of Lagrangian points for two bi-monthly periods: November-December (left) and February-March (right) for the NYF-based simulation with the GX1V6 grid. The tracks during the second year of the Lagrangian tracking are shown.

We agree with the comments on using the Lagrangian tracking for the study of sea ice transport through localized distribution of points. We have investigated the transport and loss of Lagrangian points in Sec. 3.1, showing 30.1% of all points are lost through Fram Strait export, while others are lost within the basin due to melting. Further analyses of Lagrangian points' drift, especially with densely deployed points, are planned for future study with high-resolution and historical simulations.

*3. The authors compare the model track with buoy track (figure 7) and evaluate differences between the model points and the corresponding buoys (figure 8). While three possible reasons for the tracking uncertainties are outlined, additional details on each factor would enhance the understanding of their impact. For example, the authors could explore the effects of spatial resolution by conducting sensitivity analyses with varying resolutions and comparing results for different cases. The authors could try to quantify the effects of uncertainty in atmospheric forcing on tracking, if possible. Can you also increase the size of '+' markers on the map? It is hard to see in the printed version.*

**Reply**: we agree with the referee for pointing out the 3 major contributing factors, which we list here for discussion: (1) limited model/grid resolution; (2) initial displacement between physical buoys and the nearest Lagrangian points; (3) the forcing's spatial & temporal resolution and model's inherent uncertainty (to simulate the drift). In general, we totally agree with the referee's suggestion for potential directions for analysis.

First, the model's resolution we use for the historical run (over 60 years in length) is nominally 1 degree. We plan to use LKF-capable resolutions for a historical simulation. At the coarsest, the TS015 grid (7km in the Arctic) should be used. Preferably, the 2.4km-resolution TS005 grid is much finer for deformation fields (Xu et al., 2021). However, such simulations incur much longer simulations, which is a compromise we had to make during this whole work was carried out. We are also actively gathering resources to carry out such a multi-decadal run with LKF resolving resolution in the near future.

Second, the exact matching between the buoys' initial location and deployment time and those of the Lagrangian points is planned in the second version of the Lagrangian tracking in CICE. Other collateral updates include a namelist-based configuration of the tracking module, as well as the implementation in CICE6. With these supports, the sensitivity of tracking results to the initial locations' mismatch can be examined in a systematic way.

Third, the IAF forcing is from NCEP and CORE2 datasets, which we plan to update with the JRA55-do dataset for simulating sea ice. JRA55 has a much higher spatial resolution (>4x). A comparative study between JRA55-do-based and CORE2-based historical experiments is planned, focusing on sea ice tracking. Specifically, we plan to utilize the coupling framework of CESM2 to carry out the new experiments.

Besides, according to the suggestion on the readability of the figure, we have increased the size of '+' markers in Fig. 8, as well as the overall size of the map.

*4. The authors used convex structure functions to fit sea ice deformation trends. While referencing observed multi-fractal deformations is valuable, further discussion on the selection of this specific form of function would enhance clarity.*

**Reply**: as suggested by the referee, we made revisions by adding the following sentence to the manuscript: "A generalized analysis framework with non-fixed degree of multifractality for the sea ice formations is also available (Weiss 2008; Bouchat et al.,

2022). For comparison, the forms in Eqs. 6 and 7 also assume the underlying multi-fractal, long-normal multiplicative model. In this study they are adopted, because their quadratic form is sufficient in capturing the convex shape of the structure functions (Marsan et al., 2004; Rampal et al., 2019)".

5. *In the discussion of spatial scaling around Dec. 20th and Feb. 6th for four different temporal scales (figures 10 and 11), the 3-day and 10-day cases exhibit a larger difference in beta compared to the 1-day and 30-day cases for both dates. Could you provide the reason for this?*

**Reply**: we agree with the referee's comment on the spatial scaling properties at different temporal scales around Dec-20 and Feb-6. The structure functions of β are shown in the figure below (segmented from Fig. 10 and Fig. 11), with the first (second) row showing the results for days around Dec-20 (Feb-6).

[Figure]

Several contributing factors could lead to the differences in the β function, including the convexity parameter ($a$ in Eqs. 6) which decreases more evidently between 10-day and 30-day scale for Dec-20, compared to that of Feb-6, for which $a$ decreases more between 3-day to 10-day scale. We conjecture that the different weather processes during the two periods are the major reason. Below is the Arctic Oscillation (AO) index of the NYF dataset which characterizes the large-scale atmospheric circulation and hence the sea ice drift pattern, as well as the value of β(1) simulated with a similar CICE configuration (Xu et al., 2021).

[Figure]

As shown, the value of the structure function undergoes large changes throughout the winter (esp., note the large day-to-day change). Combined with other factors such as the sea ice parameters (thickness distribution and deformation history), much variability in the structure function is present. First and foremost, the analysis should differentiate the weather events, so that the sensitivity to both typical circulation patterns and temporal scales can be elucidated. However, this analysis is not the focus of this study, we plan to carry out the analysis with historical simulations (with more events, as well as Arctic warming) and sufficient, LKF resolving resolutions.

6. *In appendix B detailing the calculation of sea ice deformation, it is important to consider scenarios where the grid becomes highly deformed, akin to figure B1.C. I am curious whether the patch is redrawn at every time step. If not, how could sea ice deformation be calculated in the case of highly deformed grids?*

**Reply**: we would like to clarify that: Fig. B1 only shows schematics of deformation fields.

The highly deformed cases usually correspond to extreme samples in the statistics of $\dot{\epsilon}$, mainly due to the localization (or intermittency) of the deformations. However, even if the grid is highly deformed, the deformation rate can still be computed through line integral over the area covered by the same set of Lagrangian points (methods in Appendix B).

It is worth noting that, existing scaling analysis works that utilize Lagrangian points in the Arctic are usually based on simple quadrilaterals [involving 4 corner points, see Marsan et al. (2004)] or even point pairs (Rampal et al., 2008), the lack of available data being a key reason. Highly deformed cases are definitely possible, but they do not hinder the overall statistical analysis. Preferably, the cells with high deformations can also be avoided to reduce the potential uncertainty in the estimated deformation rates, as is carried out in Marsan et al. (2004). We also adopt this strategy for our analysis, as introduced in Sec. 3.3.

**References**:

Marsan, D., Stern, H., Lindsay, R., and Weiss, J. (2004), *Scale Dependence and Localization of the Deformation of Arctic Sea Ice*, Phys. Rev. Lett., 93, 178 501, https://doi.org/10.1103/PhysRevLett.93.178501, 2004.

Rampal, P., J. Weiss, D. Marsan, R. Lindsay, and H. Stern (2008), *Scaling properties of sea ice deformation from buoy dispersion analysis*, J. Geophys. Res., 113, C03002, doi:10.1029/2007JC004143.

7. *The figure numbers in lines 94 and 96 on page 4 are typos. Figure 1 should be there.*

**Reply**: corrected.

8. *In line 101, there is a typo: "the cell the point" needs correction.*

**Reply**: corrected.

---

## Author Comment (AC2)

The authors would like to thank the editor and the referee's comments on our manuscript. Following the comments, we make the following replies and corresponding revisions to the manuscript. Each item of the original comments from the referee is in *green italic*, followed by our reply. Moreover, in the marked version of the revised manuscript, the revisions are highlighted with 'REV2'.

**REVIEW 2**

*This manuscript is technically sound, offers a new code likely to be of wide interest within the sea ice modeling community, and presents compelling results to clear up an unresolved question in previous sea ice model intercomparison projects. There were a number of issues left hanging, for example, in the works of Bouchat et al. (2022) and Hutter et al. (2022) for the Sea Ice Rheology Experiment (SIREx). One of the issues was the somewhat nebulous role of the ocean in the modeled sea ice deformation statistics. That remains unresolved in this work, since a simple mixed layer ocean has been used to generate sea ice deformation statistics. The second unresolved issue was the difficulty in consistently accounting for deformation across Lagrangian and Eulerian models, which is a problem this paper addresses. The scientific results are excellent and worthy of publication. The issues I found amount to relatively minor edits and additions: 1) English in this paper needs a thorough proofing; 2) Some math notation should be aligned with existing sea ice literature; 3) I suggest editing the methods section so there can be no doubt this paper concerns a diagnostic tool rather than one influencing the physics of CICE; 4) In the presentation of results, please give consideration to a figure that would convey key climatological circulation features of the Arctic to help provide visual proof that your tool works correctly; and finally 5) perhaps most importantly for GMD, please make sure the Zenodo code base provides everything needed in CICE to reproduce results documented in this paper. I provide further details here:*

**Reply**: we sincerely thank the referee for the comment about our paper addressing the Eulerian and Lagrangian diagnostics for sea ice deformations. Regarding the first aspect of the ocean dynamics' role in modulating sea ice deformations, we also plan to carry out a detailed study on this issue in the coming future with the Lagrangian tracking framework introduced in this paper.

According to the referee's suggestions, we revise the paper in several ways. First, an overall for better language usage is carried out. Second, figures and presentation in the methods section are revised to be clear of the role and validity of the Lagrangian tracking framework. Also math notations are re-checked and revised accordingly, so that they are in accordance with what is usually in the community. Third, the whole CESM/CICE version is provided through Zenodo, which is the codebase that the newly developed Lagrangian tracking is based upon, at: https://zenodo.org/records/12200190. The manuscript is revised accordingly.

*Lexicon:*

*Here are examples of the issues I noticed in my reading with English grammar and notation:*

- *Sea ice deformation should not be pluralized (i.e. deformation not deformations)*

**Reply**: corrected.

- *Change: "Since they drift with the sea ice, their locations are also representative of the sea ice floe they are attached to" to "Since they drift with the sea ice, their locations are also representative of the sea ice floe to which they are attached". (line 30)*

**Reply**: corrected.

- *Pluralize "sea ice dynamic" to "sea ice dynamics".*

**Reply**: corrected.

- *Instead of denoting deformation rates as 's, please use the correct mathematical expression for these rates to be consistent with existing sea ice rheological literature.*

**Reply**: corrected to $\dot{\epsilon}$ for all relevant cases.

- *Change "…are capable to reproduce certain…" to "…are capable of reproducing certain…". (line 55)*

**Reply**: corrected.

- *"It is worth to note…" should be changed to "It is worth noting…" (line 155)*

**Reply**: corrected.

*Please carefully proof the entire document to address similar English grammar issues.*

**Reply**: we have carried out an overhaul of the manuscript to improve the English language use.

*Methods:*

*The methods section is well explained, and I suggest only needs minor edits:*

- *For section 2.2.1, please consider a section title more useful than "Basic Support". "Software implementation" may be more descriptive.*

**Reply**: changed to "Software implementation".

- *In Figure 3, the "Lagrangian Tracking" box would suggest that somehow the tracking is instrumental in the physics of your version of CICE, when it's actually a diagnostic but not prognostic tool. I suggest the "Lagrangian Tracking" box should be shifted to the side of and outside of the step_dynamics box so that there can be no confusion on this point.*

**Reply**: we would like to clarify the current layout of the schematic plot is to show clearly when the Lagrangian tracking is carried out. According to the referee's suggestion, we move the step of Lagrangian tracking outside the step_dynamics box to be more clear about the diagnostic nature of the Lagrangian tracking in the model. The updated figure is shown below:

[Figure]

*Results:*

*There are a number of issues that should be addressed in the graphical presentation of your results:*

- *The color scale in Figure 5 is smooth to the point of deception. It appears that there is a significant problem with your thickness field in that you have near zero thickness in patches surrounding the North Pole. I may have missed it, but you appear to have omitted any comment about this problem. The problem does not preclude publication, and is almost certainly an artifact of your forcing, but please comment on the cause of this.*

**Reply**: we thank the referee for pointing out this issue, and we would like to make the following clarifications over the numerical experiments and corresponding revisions:

First, the modeled sea ice thickness field is highly dependent on the sea ice strength parameterization scheme. The R75 scheme (Rothrock, 1975) is usually shown to simulate poorer sea ice climatology than H79 (Hibler, 1979), although R75 is arguably more physics-based. Both schemes are provided by CICE and used by various studies, and the comparison between them also reported by many [such as Ungermann et al. (2017)], and by our previous study as well [Xu et al. (2021), see also the figure below]. In this study, we choose R75 mainly because it relates the ice strength to the amount of work during ridging, despite the fact that it usually performs poorer than H79 in terms of modeled climatology. One particular apparent shortcoming of R75 in the NYF-based experiments is the hoarding of thick ice in the Beaufort Gyre system and too thin ice in the Nansen Basin (as shown in our Fig. 5 and pointed out by the referee). We consider that this is due to a series of complex issues, including the fact that the CICE model cannot melt away thick, deformed ice effectively as observed (Evgenii et al, 2023). This issue is definitely worth further investigating, especially the role of external factors (i.e., forcing and coupled process) as well as internal processes to CICE (i.e., strength parameterization, etc).

[Figure]

Figure. Quasi-equilibrium March sea ice thickness with R75 (left) and H79 (right) ice strength parameterization in the CICE simulations based on NYF and slab-ocean. All other model configurations are kept the same. Hoarding of thick ice in the Beaufort Sea is apparent in the run with R75.

Second, although the model simulates different sea ice thickness fields with R75 and H79, the wintertime sea ice extent are both consistent with the satellite-based climatology. During summer, the difference is more pronounced and apparent, due to the sea ice loss in the Atlantic sector and the relatively thinner ice to start with from the

beginning of the melt season. We want to emphasize that: the sea ice drift is mainly driven by atmospheric forcing (i.e., winds), and the result of this study does depend on the sea ice strength parameterization. However, which strength scheme simulates better sea ice deformation is beyond the scope of this study, since the introduction of Lagrangian tracking is our focus here.

Third, regarding the suggestion to improve the figure's readability, we have revised the colormap to make it more readable across all thickness ranges. We apologize for the inconvenience that might have been caused, and we would like to clarify that it was not chosen on purpose.

**References**:

Hibler, W. D.: *A Dynamic Thermodynamic Sea Ice Model*, J. Phys. Oceanogr., 9, 815–846, https://doi.org/10.1175/15200485(1979)009<0815:ADTSIM>2.0.CO;2, 1979.

Rothrock, D. A.: *The energetics of the plastic deformation of pack ice by ridging*, J. Geophys. Res., 80, 4514–4519, https://doi.org/10.1029/JC080i033p04514, 1975.

Ungermann, M., Tremblay, L. B., Martin, T., and Losch, M.: *Impact of the ice strength formulation on the performance of a sea ice thickness distribution model in the Arctic*, J. Geophys. Res.-Oceans, 122, 2090–2107, https://doi.org/10.1002/2016JC012128, 2017.

Xu, Shiming, Jialiang Ma, Lu Zhou, Yan Zhang, Jiping Liu, and Bin Wang: *Comparison of sea ice kinematics at different resolutions modeled with a grid hierarchy in the Community Earth System Model (version 1.2.1)*, Geosci. Model Dev., 14, 603–628, https://doi.org/10.5194/gmd-14-603-2021, 2021.

- *You have a wonderful Lagrangian tracking tool, but at no point in the manuscript do you make use of it to show streamlines exceeding two weeks in length. Rather than just a novelty, it is essential to show at least seasonal tracks for proof that your tracking system is able to perceive the most basic of circulation features of the Arctic: the Beaufort Gyre and the Transpolar drift. I recommend including a separate figure accompanying Figure 6 with seasonal breakdowns of a less dense sub-selection of the tracer points in that figure. Without this, it is very difficult to make sense of, for example, Figure 6(b).*

**Reply**: according to the referee's comment, we have included another figure after Figure 6 (now the new Figure 7). This figure shows two-months SLP and Lagrangian points' tracks for both November-December and February-March. Data are from the same experiment (specifically, the second year of the G16 simulation under NYF). This new figure (shown below) should establish the concept that the Lagrangian tracking is working as intended.

[Figure]

Figure. Sea-level pressure (SLP) and the tracks of Lagrangian points for two bi-monthly periods: November-December (left) and February-March (right) for the NYF-based simulation with the GX1V6 grid. The tracks during the second year of the Lagrangian tracking are shown.

*Code and Model Specifics relevant to GMD:*

*Information provided in Appendix A offers far from an exhaustive list of model parameters in CICE, and it's difficult to understand why only some default constants and parameters are listed. Either make the table exhaustive of the model parameter space, or only show parameters you have changed from the default CESM version 2 code base, and state so. Please ensure, that a namelist for the model is available, or make it clear that the CICE namelist remains unchanged from CESM.*

**Reply**: we revise the manuscript to include a more exhaustive list of relevant model parameters. Also, a namelist is provided as a supplementary file for the GX1V6 experiment.

*It appears that in the Zenodo link you have provided, only the subroutines needed to be switched out in the CESM code are given. However, the CESM code is a GitHub release, and therefore subject to change. My understanding is that GMD, like other journals, now requires a stable and orphaned code base rather than a link subject to institutional change. As your Zenodo link currently stands, one could not take that code and preproduce the results in this paper without reliance on NCAR. That needs to be fixed.*

**Reply**: according to the suggestion from the referee, we have archived the CESM codebase on Zenodo, alongside the new code we have added. The link is publicly

available at: https://zenodo.org/records/12200190. The manuscript is also revised to include the updated link.

*References*

*Bouchat, A., Hutter, N., Chanut, J., Dupont, F., Dukhovskoy, D., Garric, G., et al. (2022). Sea Ice Rheology Experiment (SIREx): 1. Scaling and statistical properties of sea-ice deformation fields. Journal of Geophysical Research: Oceans, 127, e2021JC017667. https://doi.org/10.1029/2021JC017667*

*Hutter, N., Bouchat, A., Dupont, F., Dukhovskoy, D., Koldunov, N., Lee, Y. J., et al. (2022). Sea Ice Rheology Experiment (SIREx): 2. Evaluating linear kinematic features in high-resolution sea ice simulations. Journal of Geophysical Research: Oceans, 127, e2021JC017666. https://doi.org/10.1029/2021JC017666*

**Reply**: these 2 references are already cited at relevant locations of the manuscript.